# A Sample-Efficient Conditional Independence Test
# in the Presence of Discretization

**Boyang Sun** [1]  **Yu Yao** [2]  **Xinshuai Dong** [3]  **Zongfang Liu** [1]  **Tongliang Liu** [2 1]  **Yumou Qiu** [4]  **Kun Zhang** [3 1]

## Abstract

In many real-world scenarios, interested variables are often represented as discretized values due to measurement limitations. Applying Conditional Independence (CI) tests directly to such discretized data, however, can lead to incorrect conclusions. To address this, recent advancements have sought to infer the correct CI relationship between the latent variables through binarizing observed data. However, this process inevitably results in a loss of information, which degrades the test's performance. Motivated by this, this paper introduces a sample-efficient CI test that does not rely on the binarization process. We find that the independence relationships of latent continuous variables can be established by addressing an *over-identifying* restriction problem with *Generalized Method of Moments* (GMM). Based on this insight, we derive an appropriate test statistic and establish its asymptotic distribution correctly reflecting CI by leveraging node-wise regression. Theoretical findings and Empirical results across various datasets demonstrate that the superiority and effectiveness of our proposed test. Our code implementation is provided in https://github.com/boyangaaaaa/DCT.

## 1. Introduction

Conditional independence tests for discrete variables are fundamental in statistical analysis and widely applied across various disciplines. Traditional methods including the chi-squared test (F.R.S., 2009), the G-test (likelihood ratio test) (McDonald, 2009), and measures based on conditional mutual information (Kubkowski et al., 2021) are well established and extensively used. However, a critical yet often

[1]Mohamed bin Zayed University of Artificial Intelligence [2]Sydney AI Centre, The University of Sydney [3]Carnegie Mellon University [4]Peking University. Correspondence to: Yumou Qiu <qiuyumou@math.pku.edu.cn>, Kun Zhang <kunz1@cmu.edu>.

*Proceedings of the 42nd International Conference on Machine Learning*, Vancouver, Canada. PMLR 267, 2025. Copyright 2025

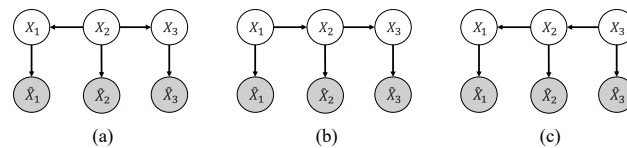

*Figure 1.* Illustration of data generative processes using causal graphical models: (a) fork, (b) and (c) chain. The discretization process maps latent continuous variables (white nodes) to observable discrete variables (gray nodes), denoted with a tilde ($\sim$).

overlooked issue is whether the analyzed variables are *truly discrete* or if they are *inherently continuous but appear discrete due to measurement limitations*.

In many real-world applications, data collection methods impose unavoidable discretization. That is, continuous variables are artificially binned into categories because of constraints in measurement precision. This phenomenon across various fields, including finance (Changsheng & Yongfeng, 2012; Damodaran, 2012), psychology (Mossman et al., 2017; Johnson et al., 2019), and recommendation systems (Sparling & Sen, 2011; Dooms et al., 2013). In these domains, inherently continuous variables—such as stock prices, cognitive ability scores, and user preferences—are frequently transformed into discrete scales, often leading to biases in statistical inference.

When discretization occurs, traditional CI tests can fail to capture the true CI relationship. As shown in Figure 1 where we illustrate the discretization process with a causal graphic model (Pearl, 2000), for all CI tests unaware of discretization, the intent is to test the CI of latent continuous variables $X_1, X_3$ given $X_2$, what is actually being tested is their discretized counterparts $\tilde{X}_1, \tilde{X}_3$ given $\tilde{X}_2$. According to the faithfulness assumption (Spirtes et al., 2000), we can infer that $X_1 \perp\!\!\!\perp X_3 | \{X_2\}$ while $\tilde{X}_1 \not\perp\!\!\!\perp \tilde{X}_3 | \{\tilde{X}_2\}$. This mismatch between the latent continuous variables and their discretized counterparts causes traditional CI tests, when applied to discretized observations, to draw incorrect conclusions about the true CI relationships of interested variables.

Recent work Discretization-Aware CI test (DCT) (Sun et al.,

2024) has attempted to address this issue by establishing the correct relationship between discretized data and latent variables through binarization of the observed data. While this approach facilitates more accurate CI testing by simplifying the data structure, it inherently leads to a loss of information. The reduction of data to binary form can significantly degrade the performance of CI tests, especially in settings with small sample sizes where the preservation of information is crucial for reliable statistical inference.

Motivated by the limitations of existing methodologies, this paper aims to introduce a sample-efficient CI test that circumvents the need for binarization, thereby preserving the full richness of the data. Our approach leverages the Generalized Method of Moments (GMM) to address the over-identifying restrictions problem, enabling the estimation of covariance of latent continuous variables without sacrificing information. We then adopt the strategy of DCT to derive test statistics and their asymptotic distribution for CI testing, utilizing nodewise regression (Callot et al., 2019). The paper seeks to contribute as follows:

- We propose Discretization-Aware CI Test with GMM (DCT-GMM), a novel CI test tailored for discretization.

- We provide a theoretical analysis proving that DCT-GMM is consistent and has lower variance than DCT, making it more sample efficient. (Ziegel, 2002).

- We empirically demonstrate DCT-GMM's effectiveness and superiority over state-of-the-art CI tests, particularly in small-sample regimes.

## 2. Related Work

**Conditional Independence Test**   Testing for CI is a fundamental concept in statistics, with linear Gaussian models traditionally dominating due to their simplicity and interperability. These models assume linear dependencies and Gaussian noise, providing closed-form solutions for testing through metrics like partial correlation (Yuan & Lin, 2007; Peterson et al., 2015; Mohan et al., 2012; Ren et al., 2015). However, the linear Gaussian assumption restricts the generality. Recent CI testing advancements leverage kernel methods for nonlinear continuous relationships (Fukumizu et al., 2004). Methods like KCI (Zhang et al., 2012) and RCI (Strobl et al., 2019) analyze partial associations, while KCIP (Doran et al., 2014) employs sample permutations to simulate CI. For discrete variables, $G^2$ (Aliferis et al., 2010) and conditional mutual information (Zhang et al., 2010) are standard tests. A recent advance on permutation-based rank test, MPRT (Dong et al., 2025), can also be used to test vanishing partial correlation in the presence of discretization.

**Prior work**   DCT (Sun et al., 2024) moves the first step towards the CI test specifically for the discretization scenario.

Their approach can be decomposed into three steps: 1. The calculation of estimated covariance $\hat{\boldsymbol{\Sigma}}$, based on the property that the proportion of both observed variables exceeding their means reflects the underlying covariance, solved using *a single equation*. 2. The deviation of covariance matrix $\hat{\boldsymbol{\Sigma}} - \boldsymbol{\Sigma}^*$ follows a multivariate normal distribution utilizing Z-estimator(Vaart, 1998). 3. The deviation of precision matrix $\hat{\boldsymbol{\Omega}} - \boldsymbol{\Omega}^*$ also follows a multivariate normal distribution utilizing node-wise regression (Callot et al., 2019). However, despite having *multiple solvable equations* from the discretized observations, only one parameter of interest exists per variable pair, leading to an over-identification issue. Efficiently utilizing all available information is the key challenge, motivating us to explore the usage of GMM.

**Generalized Method of Moments**   GMM (Newey, 2007; Hansen, 1982) is a statistical estimation technique offering a principal solution to the over-identification problem. Suppose $\boldsymbol{\theta}$ denotes a $p \times 1$ vector, $x^i$ denotes the observation of a data sample where $i \in (1, \ldots, n)$ is the index. $f_i(\boldsymbol{\theta}) = f(x^i, \boldsymbol{\theta})$ be a $m \times 1$ vector of functions. For the true parameter $\boldsymbol{\theta}^*$, we will have

$$\mathbb{E}[f_i(\boldsymbol{\theta}^*)] = \mathbf{0}.$$

Using $\hat{g}(\boldsymbol{\theta}) = \frac{1}{n} \sum_{i=1}^{n} f_i(\boldsymbol{\theta})$ denote the sample average of the $f_i(\boldsymbol{\theta})$. The $\mathbf{A}$ is a $m \times m$ positive semi-definite matrix. The GMM estimator is given by

$$\hat{\boldsymbol{\theta}} = \arg\min_{\boldsymbol{\theta}} \quad \hat{g}(\boldsymbol{\theta})^T \mathbf{A} \hat{g}(\boldsymbol{\theta}),$$

providing a framework for valid inference (Newey, 2007). In our case, by properly designing the moment functions and selecting the parameter of interest, GMM can efficiently fulfill the objective of estimation and addressing the over-identification issue.

## 3. DCT-GMM: Discretization-Aware CI Test with GMM

**Notation**   Throughout this work, we use $X_j$ to denote the $j$-th component of the vector of variables $\boldsymbol{X} = (X_1, \ldots, X_p)$ with finite observations $\{x_j^1, \ldots, x_j^n\}$. We denote the sample mean given $n$ samples by $\mathbb{E}_n[X_j] = \frac{1}{n} \sum_{i=1}^{n} x_j^i$ while its true expectation is $\mathbb{E}[X_j]$. Similarly, the empirical probability is represented by $\mathbb{P}_n$, and the true probability by $\mathbb{P}$. For a parameter $\alpha$, its true value is $\alpha^*$ and its estimation is $\hat{\alpha}$. For a matrix $\mathbf{X}$, $\mathbf{X}^{-T}$ denotes the transpose of its inverse. We use $\mathbf{X}_{-j}$ to represent all other columns of $\mathbf{X}$ without $X_j$. Similarly, $\mathbf{X}_{-j-j}$ is the submatrix of $\mathbf{X}$ without $j$th column and $j$th row, and the $\mathbf{X}_{-jj}$ is the vector of $j$th column without $j$th row. For a full notation table, please refer to Appendix A.1.

**Problem Setting**   In this paper, we adopt the same nonparanormal model as DCT (Sun et al., 2024). Specifically,

we consider a set of independent identically distributed (i.i.d.) $p$-dimensional random discrete variables, denoted as $\tilde{\boldsymbol{X}} = (\tilde{X}_1, \tilde{X}_2, \ldots, \tilde{X}_j, \ldots, \tilde{X}_p)$. For each discrete variable $\tilde{X}_j$ with finite observations $\{\tilde{x}_j^1, \ldots, \tilde{x}_j^n\}$, there exists a corresponding latent Gaussian variable $X_j$. The transformation from $X_j$ to $\tilde{X}_j$ is governed by an unknown monotone nonlinear function $g_j$ and a thresholding function $f_j$. The function $f_j \circ g_j : \mathcal{X} \rightarrow \tilde{\mathcal{X}}$ maps the continuous domain of $X_j$ onto the discrete domain $\tilde{\mathcal{X}}_j$. Specifically, for each variable $X_j$, there exists a finite constant vector $\mathbf{d}_j = [d_{j,1}, \ldots, d_{j,M-1}]$ characterized by strictly increasing elements such that

$$\tilde{X}_j = f_j(g_j(X_j)) = \begin{cases} 1 & g_j(X_j) < d_{j,1}, \\ m & d_{j,m-1} < g_j(X_j) < d_{j,m}, \\ M & g_j(X_j) > d_{j,M-1}. \end{cases} \quad (1)$$

Equivalently, we can conclude $\tilde{X}_j = m$, for $g_j^{-1}(d_{j,m-1}) < X_j < g_j^{-1}(d_{j,m})$, where $m$ is an integer ranging from 1 to $M$. That is, there exists a finite constant vector $\mathbf{c}_j = [g_j^{-1}(d_{j,1}), \ldots, g_j^{-1}(d_{j,M-1})]$ acting as the "discretization boundary" that partition the $X_j$ into $M$ categories. We refer $M$ as the cardinality of the discrete variable $\tilde{X}_j$. Without loss of generality, we assume $\boldsymbol{X} \sim N(\mathbf{0}, \boldsymbol{\Sigma})$ with $\boldsymbol{\Sigma} = (\sigma_{j_1 j_2})$ and $\sigma_{jj} = 1$. That is, we assume the original continuous variables $\boldsymbol{X}$ follow a multivariate normal distribution with zero mean and unit variance on the diagonal of $\boldsymbol{\Sigma}$. We provide a detailed discussion regarding its rationality and w.l.o.g in Appendix B.

**Objective** We aim to develop a CI test to infer the correct CI relationship between latent continuous variables $\boldsymbol{X} = (X_1, \ldots, X_p)$, which are the interested ones given their discretized observations $\tilde{\mathbf{X}}$ only. By assuming a linear Gaussian of original continuous variables, our objective directly transfers to deduce the statistical inference of covariance matrix $\boldsymbol{\Sigma} = (\sigma_{j_1 j_2})$ for independent test and precision matrix $\boldsymbol{\Omega} = \boldsymbol{\Sigma}^{-1} = (\omega_{jk})$ for CI test (Baba et al., 2004). Specifically, the covariance $\sigma_{j_1 j_2} = 0$ indicates that $X_{j_1} \perp\!\!\!\perp X_{j_2}$, and the precision coefficient $\omega_{jk} = 0$ indicates that $X_j \perp\!\!\!\perp X_k | \boldsymbol{X}_{-\{jk\}}$, where $\boldsymbol{X}_{-\{jk\}}$ represents all other variables in $\boldsymbol{X}$ except $X_j$ and $X_k$. Technically, we are interested in two key tasks:

- Estimation: Obtain $\hat{\sigma}_{j_1 j_2}$ and $\hat{\omega}_{jk}$ serving as the estimation of the corresponding true parameters with only discretized observations available.

- Inference: Derive the distribution $\hat{\sigma}_{j_1 j_2} - \sigma_{j_1 j_2}^*$ for independence test and $\hat{\omega}_{jk} - \omega_{jk}^*$ for CI test.

In the subsequent section, we develop our theoretical framework through three key steps. First, we demonstrate that for any pair of continuous variables $X_{j_1}$ and $X_{j_2}$ with their corresponding discretized observations $\tilde{X}_{j_1}$ and $\tilde{X}_{j_2}$, we can

effectively construct both the estimator $\hat{\sigma}_{j_1 j_2}$ and characterize the distribution of $\hat{\sigma}_{j_1 j_2} - \sigma_{j_1 j_2}^*$ using GMM. Second, we establish that the nodewise regression parameter $\beta_{j,k}$ serves as an effective surrogate for the precision matrix element $\omega_{jk}$. Finally, we show the asymptotic normal distribution of $\hat{\beta}_{j,k} - \beta_{j,k}^*$ by analyzing its relationship with the component distributions of $\hat{\sigma}_{j_1 j_2} - \sigma_{j_1 j_2}^*$.

### 3.1. GMM for covariance estimation and inference

**Estimating discretization boundaries** The discretization scheme maps the $X_j$ onto a finite set of discrete values according to the discretization boundaries, maintaining the ordinal relationship of the original continuous variable while reducing its resolution. For ease of notation, we denote the augmented discretization boundary $\mathbf{c}_j^* := [c_{j,0}^*, c_{j,1}^*, \ldots, c_{j,M-1}^*, c_{j,M}^*] = [-\infty, g_j^{-1}(d_{j,1}), \ldots, g_j^{-1}(d_{j,M-1}), +\infty]$. We further denote $\Phi(\cdot)$ as the cumulative distribution function (cdf) of the standard normal distribution. Our available observation consists of binned discrete values. Since $X_j \sim N(0,1)$ according to the assumption, we conclude that $\mathbb{P}(\tilde{X}_j = m) = \mathbb{P}(c_{j,m-1}^* < X_j < c_{j,m}^*)$. That is, the probability of observing a discrete value corresponds to the probability of the original continuous variable falling into a particular region. Although the true probability is not directly accessible, it can be estimated by calculating the sample proportion of the observations within each bin. Specifically, we can obtain the estimation of the discretization boundaries

$$\hat{c}_{j,m} = \Phi^{-1}(\sum_{k=1}^{m} \hat{\tau}_{j,k}), \quad (2)$$

where $\hat{\tau}_{j,k}$ is the empirical probability defined as $\hat{\tau}_{j,k} := \mathbb{P}_n(\tilde{X}_j = k) = \frac{1}{n} \sum_{i=1}^{n} \mathbb{1}(\tilde{x}_j^i = k)$, serving as the estimation of the true probability $\tau_{j,k} := \mathbb{P}(\tilde{X}_j = k)$. The indicator function $\mathbb{1}(condition)$ is 1 if the condition hold true, 0 otherwise. This formulation provides a closed-form solution for estimating the discretization boundaries from observed discrete data.

**Estimate covariance through single equation** The challenge lies in estimating the latent covariance $\sigma_{j_1 j_2}$ with discretized values. For a pair of continuous variables $X_{j_1}$ and $X_{j_2}$, the discretization scheme essentially creates a "grid". Each cell represents a specific combination of discretized values for both variables. When we count how many samples fall into one cell, the sample proportion within each cell provides an empirical estimate of the joint probability density, which can be expressed as $\mathbb{P}_n(\tilde{X}_{j_1} = m, \tilde{X}_{j_2} = k)$, serving as the estimation of the true probability $\mathbb{P}(c_{j_1,m-1}^* < X_{j_1} < c_{j_1,m}^*, c_{j_2,k-1}^* < X_{j_2} < c_{j_2,k}^*)$.

According to our assumption that the latent variables follow a multivariate normal distribution, the true probability

above is given by $\Phi(c^*_{j_1,m-1}, c^*_{j_1,m}, c^*_{j_2,k-1}, c^*_{j_2,k}; \sigma^*_{j_1j_2})$, which is the cdf of a bivariate normal distribution with the true covariance $\sigma^*_{j_1j_2}$ integrated over the rectangular region defined by $[c^*_{j_1,m-1}, c^*_{j_1,m}] \times [c^*_{j_2,k-1}, c^*_{j_2,k}]$. For a specific form of the function, please refer to Appendix A.2.

For notational convenience, we define $\hat{\tau}_{j_1j_2,mk} := \mathbb{P}_n(\tilde{X}_{j_1} = m, \tilde{X}_{j_2} = k)$ as the empirical joint probability, and $\tau_{j_1j_2,mk} := \mathbb{P}(\tilde{X}_{j_1} = m, \tilde{X}_{j_2} = k)$ as the true probability. We use $\hat{\tau}^i_{j_1j_2,mk} = \mathbb{1}(\tilde{x}^i_{j_1} = m, \tilde{x}^i_{j_2} = k)$ as the indicator of sample $i$. The empirical cell density can be easily computed from the observation as $\hat{\tau}_{j_1j_2,mk} = \frac{1}{n}\sum_{i=1}^n \hat{\tau}^i_{j_1j_2,mk}$. The estimated covariance $\hat{\sigma}_{j_1j_2}$ can then be obtained by solving following equation:

$$\hat{\tau}_{j_1j_2,mk} = \Phi(\hat{c}_{j_1,m-1}, \hat{c}_{j_1,m}, \hat{c}_{j_2,k-1}, \hat{c}_{j_2,k}; \sigma_{j_1j_2}), \quad (3)$$

where $\hat{c}$ can be computed using Eq. (2). We call the equation above a "bridge equation", which provides a direct solution to recovering the underlying covariance by only using the discretized observations.

However, this formulation presents an overidentification challenge. For any pair of discrete variables $\tilde{X}_{j_1}$ and $\tilde{X}_{j_2}$ with cardinalities $M$ and $K$ respectively, we obtain $M \times K$ distinct cells, each corresponding to its own equation. This results in an overdetermined system with $M \times K$ equations but only one parameter of interest $\sigma_{j_1j_2}$. This overidentification is a key limitation of DCT (Sun et al., 2024), which utilizes only a single equation despite the availability of multiple informative constraints. In the following section, we demonstrate how GMM acts as a principled framework for efficiently leveraging all available information from these multiple equations, thereby offering a preciser solution.

**Move from single equation to multiple equation** For a pair of variables $\tilde{X}_{j_1}$ and $\tilde{X}_{j_2}$ with cardinality $M$ and $K$ correspondingly, we define the parameters of interest $\boldsymbol{\theta} = (\sigma_{j_1j_2}, \mathbf{c}_{j_1}, \mathbf{c}_{j_2}) \in \mathbb{R}^{M+K-1}$. Let $f_i(\boldsymbol{\theta}) = f(\tilde{x}^i_{j_1}, \tilde{x}^i_{j_2}, \boldsymbol{\theta}) \in \mathbb{R}^{MK}$ referred as the moment function with the form:

$$f_i(\boldsymbol{\theta}) = \begin{pmatrix} \hat{\tau}^i_{j_1j_2,11} - \Phi(c_{j_1,0}, c_{j_1,1}, c_{j_2,0}, c_{j_2,1}; \sigma_{j_1j_2}) \\ \vdots \\ \hat{\tau}^i_{j_1j_2,MK} - \Phi(c_{j_1,M-1}, c_{j_1,M}, c_{j_2,K-1}, c_{j_2,K}; \sigma_{j_1j_2}) \end{pmatrix}. \quad (4)$$

For the true parameter $\boldsymbol{\theta}^*$, the population moment condition satisfies $\mathbb{E}[f_i(\boldsymbol{\theta}^*)] = \mathbf{0}$. The detailed derivation of this condition can be found in Appendix F.1. Let the sample analogue of the moment condition be $\hat{g}(\boldsymbol{\theta}) = \frac{1}{n}\sum_{i=1}^n f_i(\boldsymbol{\theta})$. Given a positive semi-definite matrix $\mathbf{A} \in \mathbb{R}^{MK \times MK}$ as weighting matrix, the GMM estimator is given by

$$\hat{\boldsymbol{\theta}} = \arg\min_{\boldsymbol{\theta}} \quad \hat{g}(\boldsymbol{\theta})^T \mathbf{A} \hat{g}(\boldsymbol{\theta}). \quad (5)$$

This formulation leverages all $M \times K$ moment functions simultaneously to obtain the estimation $\hat{\boldsymbol{\theta}}$, efficiently utilizing the available information from the discretized observations. The estimated covariance $\hat{\sigma}_{j_1j_2}$ is nothing but the

first element of $\hat{\boldsymbol{\theta}}$. The next question is, how to construct the distribution $\hat{\sigma}_{j_1j_2} - \sigma^*_{j_1j_2}$. Given this, we propose the following theorem:

**Theorem 3.1.** *Under the null hypothesis that the original continuous variables $X_{j_1} \perp\!\!\!\perp X_{j_2}$, with the moment function $f_i(\boldsymbol{\theta})$ defined as Eq. (4), when number of samples $n \to +\infty$, the estimator $\hat{\sigma}_{j_1j_2}$ is asymptotically normal distributed:*

$$\sqrt{n}(\hat{\sigma}_{j_1j_2} - \sigma^*_{j_1j_2}) = -\frac{1}{n}\sum_{i=1}^n \left[ (\hat{\mathbf{G}}^T\mathbf{A}\hat{\mathbf{G}})^{-1}\hat{\mathbf{G}}^T\mathbf{A}f_i(\boldsymbol{\theta}^*) \right]_1, \quad (6)$$

*which will converge in distribution to $N(0, \mathbf{V}_{11})$,*

- *where $\mathbf{V} = (\mathbf{G}^T\mathbf{A}\mathbf{G})^{-1}\mathbf{G}^T\mathbf{A}\mathbf{S}\mathbf{A}\mathbf{G}(\mathbf{G}^T\mathbf{A}\mathbf{G})^{-1}$, and $\mathbf{V}_{11}$ is its first entry,*

- *$\mathbf{G} = \mathbb{E}[\frac{\partial f_i(\boldsymbol{\theta}^*)}{\partial \boldsymbol{\theta}^*}]$ is the expectation of the Jacobian matrix of the moment function at true parameter $\boldsymbol{\theta}^*$,*

- *$\hat{\mathbf{G}} = \mathbb{E}_n[\frac{\partial f_i(\hat{\boldsymbol{\theta}})}{\partial \hat{\boldsymbol{\theta}}}]$ is sample average of the Jacobian matrix of moment function at estimated parameter $\hat{\boldsymbol{\theta}}$,*

- *$\mathbf{S} = \mathbb{E}[f_i(\boldsymbol{\theta}^*)f_i(\boldsymbol{\theta}^*)^T]$ is the covariance matrix of the moment function $f_i(\boldsymbol{\theta}^*)$.*

The detailed derivation can be found in Appendix F.2. Since we never have the access to the true parameters $\boldsymbol{\theta}^*$ and the true expectation $\mathbb{E}[\cdot]$, in practice, we can plug in their estimation $\mathbb{E}_n[\frac{\partial f_i(\hat{\boldsymbol{\theta}})}{\partial \boldsymbol{\theta}}]$ and $\mathbb{E}_n[f_i(\hat{\boldsymbol{\theta}})f_i(\hat{\boldsymbol{\theta}})^T]$ to calculate the variance of the asymptotic distribution. For the weighting matrix $\mathbf{A}$, theoretically, any positive semi-definite matrix is applicable to the theorem above. A common choice could be the identity matrix.

**Two-step GMM** Apparently, the choice of weighting matrix $\mathbf{A}$ plays an important role in determining the statistical property of the GMM. Specifically, $\mathbf{A}$ directly influences the variance of the distribution $\hat{\sigma}_{j_1j_2} - \sigma^*_{j_1j_2}$. The question is how to choose $\mathbf{A}$ optimally to minimize the asymptotic variance of the estimator. According to the classical theory of GMM (Hansen, 1982), we have the following lemma:

**Lemma 3.2.** *Suppose the choice of $\mathbf{A} \xrightarrow{p} \mathbf{S}^{-1}$, where $\mathbf{S} = \mathbb{E}[f_i(\boldsymbol{\theta}^*)f_i(\boldsymbol{\theta}^*)^T]$ is the covariance matrix of $f_i(\boldsymbol{\theta}^*)$ for the true parameters $\boldsymbol{\theta}^*$, then*

$$\sqrt{n}(\hat{\sigma}_{j_1j_2} - \sigma^*_{j_1j_2}) = -\frac{1}{n}\sum_{i=1}^n \left[ (\hat{\mathbf{G}}^T\mathbf{A}\hat{\mathbf{G}})^{-1}\hat{\mathbf{G}}^T\mathbf{A}f_i(\boldsymbol{\theta}^*) \right]_1, \quad (7)$$

*will converge in distribution to a normal distribution with variance $N(0, \mathbf{V}_{11})$, where $V = (\mathbf{G}^T\mathbf{S}^{-1}\mathbf{G})^{-1}$, strictly smaller in the positive semi-definite sense compared to the asymptotic covariance matrix in the one-step GMM estimator given in Theorem 3.1. Here, $\mathbf{G}$ and $\hat{\mathbf{G}}$ have the same definition as in Theorem 3.1.*

The detailed derivation is provided in Appendix F.3. In practice, the procedure begins by estimating the parameter of interest using a predefined weighting matrix, such as the identity matrix. Next, the covariance of the moment functions is used to construct the optimal weighting matrix, enabling the final GMM estimator to efficiently re-estimate the parameters. This two-step GMM approach is a well-established technique for achieving asymptotic efficiency, and its superiority is empirically validated in Section 4.

The estimated covariance $\sigma_{j_1 j_2}$ serves as an indicator of unconditional independence. Following the framework proposed by (Sun et al., 2024), we build such a CI test utilizing nodewise regression.

### 3.2. Nodewise regression for constructing CI test

In this subsection, we follow (Sun et al., 2024) and leverage the nodewise regression to derive the CI test. For completeness, we present the main results here and refer to the original paper for a detailed treatment. The practical implementation is also discussed at the end of this subsection.

By assuming the $\boldsymbol{X}$ follows a multivariate normal distribution, our task of constructing CI test is equivalent to (1). computation of $\hat{\omega}_{jk}$ based on the discretized observations; (2). construction of $\hat{\omega}_{jk} - \omega_{jk}^*$. Targeting both tasks, the nodewise regression is leveraged, which shows that:

- The regression parameter $\beta_{j,k}$ can serve as an effective surrogate for testing the null hypothesis that $X_j \perp\!\!\!\perp X_k | \boldsymbol{X}_{-\{jk\}}$, i.e., $\omega_{jk} = 0$.

- The formulation of $\hat{\beta}_{j,k} - \beta_{j,k}^*$ can be expressed as a linear combination of $\hat{\sigma}_{j_1 j_2} - \sigma_{j_1 j_2}^*$, which allows us to derive its distribution and facilitating the CI test.

The following lemma establishes the key properties of nodewise regression that support this approach.

**Lemma 3.3.** *[Nodewise Regression Properties] (Sun et al., 2024) For a p-dimensional multivariate normal variable $\boldsymbol{X} = (X_1, \ldots, X_p) \sim N(0, \boldsymbol{\Sigma})$ with covariance matrix $\boldsymbol{\Sigma}$ and precision matrix $\boldsymbol{\Omega} = \boldsymbol{\Sigma}^{-1} = (\omega_{jk})_{1 \le j,k \le p}$. For any $j \in \{1, \ldots, p\}$, consider the nodewise regression where each $X_j$ is regressed on all other variables:*

$$X_j = \sum_{k \ne j} X_k \beta_{j,k} + \epsilon_j,$$

*where $\beta_{j,k}$ is the regression coefficient of $X_k$ in predicting $X_j$, $\beta_j = (\beta_{j,k})_{k \ne j} \in \mathbb{R}^{p-1}$ is the vector of all coefficients, and $\epsilon_j$ is the residual term. Then the following relationships hold:*

$$\beta_{j,k} = -\frac{\omega_{jk}}{\omega_{jj}}, \quad j \ne k.$$
$$\beta_j = \boldsymbol{\Sigma}_{-j-j}^{-1} \boldsymbol{\Sigma}_{-jj} \in \mathbb{R}^{p-1}. \tag{8}$$

The derivation can be found in Appendix F.4.1. The first row of Eq. (8) indicates that $\beta_{j,k}$ is a scaled version of $\omega_{jk}$. Since $\omega_{jj}$ will never be zero due to the positive definiteness of $\boldsymbol{\Omega}$, testing if $\beta_{j,k} = 0$ is exactly the same as testing $\omega_{jk} = 0$. Thus, $\beta_{j,k}$ serves as an effective surrogate of $\omega_{jk}$. Now our focus transfers to calculating $\hat{\beta}_{j,k}$ and deriving the distribution of $\hat{\beta}_{j,k} - \beta_{j,k}^*$.

We further note that the second row of Eq. (8) constructs a consistent relationship between $\beta_j$ and the covariance matrix $\boldsymbol{\Sigma}$. Thus, we can conduct its estimation as $\hat{\beta}_j = (\hat{\beta}_{j,k})_{j \ne k} = \hat{\boldsymbol{\Sigma}}_{-j-j}^{-1} \hat{\boldsymbol{\Sigma}}_{-jj}$, where the estimated covariance terms can be obtained through solving Eq. (5).

**Statistical Inference for $\beta_{j,k}$** Nodewise regression transfers the parameter of interest from $\omega_{jk}$ to $\beta_{j,k}$. While the estimation of $\beta_{j,k}$ has been effectively solved, the next question is how to construct the distribution of $\hat{\beta}_{j,k} - \beta_{j,k}^*$. Fortunately, we have already established the asymptotic distribution of $\hat{\sigma}_{j_1 j_2} - \sigma_{j_1 j_2}^*$. Therefore, if we can express $\hat{\beta}_j - \beta_j^*$ as a linear combination of $\hat{\sigma}_{j_1 j_2} - \sigma_{j_1 j_2}^*$, the problem is readily solved, as $\hat{\beta}_j - \beta_j^*$ will be a linear combination of dependent asymptotically normal random variables. The underlying relationship between these variables is as follows:

$$\hat{\beta}_j - \beta_j^* = -\hat{\boldsymbol{\Sigma}}_{-j-j}^{-1} \left( (\hat{\boldsymbol{\Sigma}}_{-j-j} - \boldsymbol{\Sigma}_{-j-j}^*) \beta_j^* - (\hat{\boldsymbol{\Sigma}}_{-jj} - \boldsymbol{\Sigma}_{-jj}^*) \right). \tag{9}$$

The derivation of this result is provided in Appendix F.4.2. For notational convenience, we express the difference between the estimated and true covariances as:

$$\hat{\sigma}_{j_1 j_2} - \sigma_{j_1 j_2}^* = \frac{1}{n} \sum_{i=1}^{n} \xi_{j_1 j_2}^i, \tag{10}$$

where the specific form of $\xi_{j_1 j_2}^i$ is given in Theorem 3.1 and Lemma 3.2. We further denote $\hat{\boldsymbol{\Sigma}}_{-j-j} - \boldsymbol{\Sigma}_{-j-j} = \frac{1}{n} \sum_{i=1}^{n} \boldsymbol{\Xi}_{-j-j}^i$ and $\hat{\boldsymbol{\Sigma}}_{-jj} - \boldsymbol{\Sigma}_{-jj} = \frac{1}{n} \sum_{i=1}^{n} \boldsymbol{\Xi}_{-jj}^i$ are matrix form of $\xi_{j_1 j_2}^i$.

**Conditional Independence Test** We apply the following theorem for the CI test, with proof provided at Appendix F.4.2 for completeness:

**Theorem 3.4.** *(Sun et al., 2024) Under the null hypothesis that $X_j$ and $X_k$ are conditional statistically independent given a set of variables $\boldsymbol{X}_{\{-jk\}}$, i.e., $\beta_{j,k} = 0$, the statistic*

$$\hat{\beta}_{j,k} = (\hat{\boldsymbol{\Sigma}}_{-j-j}^{-1} \hat{\boldsymbol{\Sigma}}_{-jj})_{[k]}, \tag{11}$$

*where $[k]$ denotes the element corresponding to the variable $X_k$ in $\hat{\boldsymbol{\Sigma}}_{-j-j}^{-1} \hat{\boldsymbol{\Sigma}}_{-jj}$, has the asymptotic distribution:*

$$\sqrt{n}(\hat{\beta}_{j,k} - \beta_{j,k}^*) \xrightarrow{d} N(0, \mathbf{V}), \text{ where}$$

- $\mathbf{V} = \boldsymbol{a}^{[k]^T} \frac{1}{n} \sum_{i=1}^{n} vec(\boldsymbol{B}_{-j}^i) vec(\boldsymbol{B}_{-j}^i)^T \boldsymbol{a}^{[k]},$

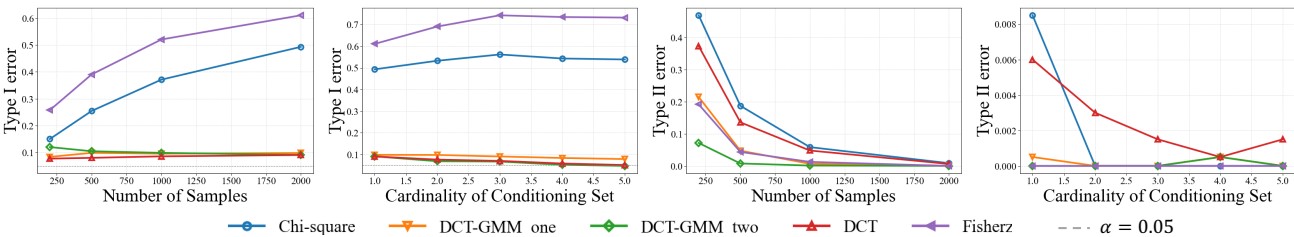

*Figure 2.* Comparison of results of Type I and Type II error (1-power) for discretized observations. DCT-GMM_one uses one-step GMM with **A** setting as identity, and DCT-GMM_two uses two-step GMM with **A** setting as the sample covariance of moment functions.

- $\mathbf{B}^i = \begin{bmatrix} \mathbf{\Xi}^i_{-jj}^T \\ \mathbf{\Xi}^i_{-j-j} \end{bmatrix}$, *and $\tilde{\beta}_j$ is $\beta_j^*$ whose $\beta_{j,k}^* = 0$,*

- $\boldsymbol{a}^{[k]} = \begin{bmatrix} -(\hat{\mathbf{\Sigma}}^{-1}_{-j-j})^T_{[k],:} \\ vec\left((\hat{\mathbf{\Sigma}}^{-1}_{-j-j})^T_{[k],:}\tilde{\boldsymbol{\beta}}_j^T\right) \end{bmatrix}$, *and vec is row-wise*

  *vectorization of a matrix, and $(\hat{\mathbf{\Sigma}}^{-1}_{-j,-j})_{[k],:}$ denotes the row in $\hat{\mathbf{\Sigma}}^{-1}_{-j,-j}$ that corresponds to $X_k$.*

**Practical implementation for DCT-GMM** In practical implementation, we estimate the regression parameter $\hat{\beta}_j$ and set $\hat{\beta}_{j,k} = 0$ as a substitute for $\tilde{\beta}_j$ to calculate variance and conduct the confidence interval test. Specifically, we derive $\hat{\beta}_{j,k}$ using the estimation equation in Equation (11), where estimated covariance terms are calculated utilizing GMM. As discussed in Section 3.1, we consider two GMM estimators: naive GMM (also called one-step GMM, the one without carefully designing **A**) and two-step GMM. In this paper, we empirically validate the effectiveness of both, whereas **A** is chosen as an identity matrix for the one-step GMM and the $\mathbb{E}_n[f_i(\hat{\boldsymbol{\theta}})f_i(\hat{\boldsymbol{\theta}})^T]$ for the two-step GMM following Lemma 3.2. The pseudo-code of both approaches is provided in Appendix C.

Under the null hypothesis of conditional independence ($\beta_{j,k} = 0$), we substitute the calculated $\hat{\beta}_{j,k}$ into the distribution defined in Theorem 3.4 to obtain the p-value. Statistical inference follows a standard hypothesis testing approach: if the p-value is less than the predefined significance level $\alpha$ (typically 0.05), we conclude that the tested pairs are conditionally dependent. Conversely, if the p-value exceeds $\alpha$, we fail to reject the null hypothesis and deduce the tested pairs are conditionally independent.

### 3.3. Comparison with DCT

One of the motivations behind DCT-GMM is to address the insufficient data utilization in DCT. This naturally raises the question: Is DCT-GMM necessarily superior to DCT? Toward this question, we propose the following theorem:

**Theorem 3.5.** *(Informal) When $n \to +\infty$, by constructing the moment functions properly, the DCT-GMM with two-step GMM has a lower variance than DCT.*

The formal theorem and detailed derivation can be found in Appendix G. Intuitively, if we construct the moment functions the same as DCT, we will reach the estimator with the same variance. However, the GMM framework allows the introduction of additional moment functions, thereby reducing the variance. We empirically validate the theorem in Section 4.3.

## 4. Experiment

We applied the proposed test DCT-GMM to synthetic dataset to evaluate its performance compared with baselines including DCT (Sun et al., 2024), Fisher-z test (Fisher, 1921), Chi-square test (F.R.S., 2009). Specifically, we investigate its Type I and Type II error in different scenarios and its application in causal discovery. The experiments investigating its performance in denser graphs and effectiveness in real-world dataset can be found in Appendix E.

### 4.1. On the Effect of the Cardinality of Conditioning Set and the Sample Size

We conducted an experimental study to examine the behavior of Type I and Type II error probabilities under two distinct experimental designs. The first design explores the impact of sample size variation, specifically testing sample sizes of $n = (200, 500, 1000, 2000)$ while maintaining a single conditioning variable, noted as $D = 1$. In the second design, we fixed the sample size at $n = 2000$ and systematically varied the number of conditioning variables $D = (1, \ldots, 5)$. We assumed that all variables in the conditioning set are influential and affect the confidence intervals of the tested pairs. Each experimental configuration was replicated 2,000 times to ensure robust statistical analysis. We use $X$ and $Y$ to denote the tested pairs and $\mathbf{Z}$ to denote the variables being conditioned on.

To assess the accuracy of the derived asymptotic null distribution, we evaluated whether the Type I error probability aligns with the predetermined significance level $\alpha = 0.05$. We first generate $\mathbf{Z}$ as an independent multivariate normal distribution whose mean and variance are randomly sampled from a uniform distribution $U(0, 1)$. We then generate cor-

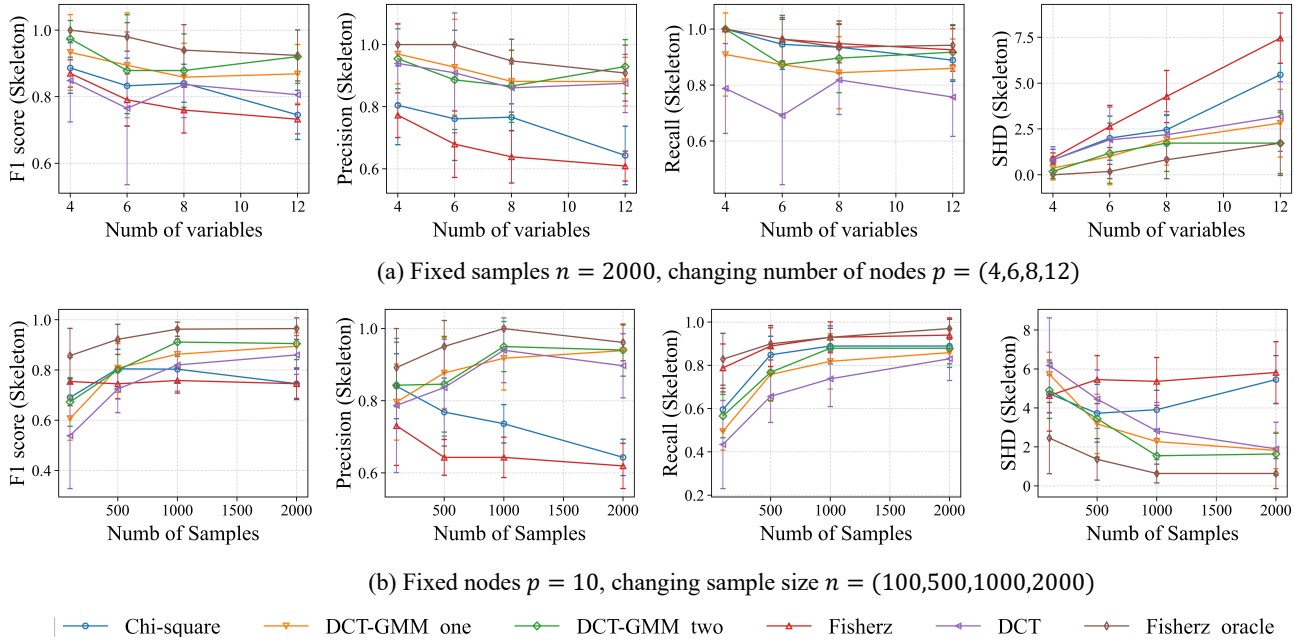

(a) Fixed samples $n = 2000$, changing number of nodes $p = (4,6,8,12)$

(b) Fixed nodes $p = 10$, changing sample size $n = (100,500,1000,2000)$

*Figure 3.* Experimental result of skeleton discovery on synthetic data for changing number of nodes (a) and changing sample size (b). Fisherz_oracle is the Fisher-z test applied to original continuous data. We evaluate $F_1$ ($\uparrow$), Precision ($\uparrow$), Recall ($\uparrow$) and SHD ($\downarrow$).

responding $X$ and $Y$ using $\mathbf{Z}$, structured as $\sum_{i=1}^{D} a_i Z_i + E_i$ (for the first scenario, $D = 1$), where $a_i$ is a scalar sampled from a standard normal distribution and $E_i$ follows a standard normal distribution. This ensures that $X \perp\!\!\!\perp Y \mid \mathbf{Z}$. The data are then discretized into three levels, with random boundaries set based on the support of each variable, producing the discretized observations $\tilde{X}, \tilde{Y}$, and $\tilde{\mathbf{Z}}$. The first two columns of Figure 2 show the resulting Type I error at a significance level of $\alpha = 0.05$.

A robust statistical test should minimize the Type II error, thereby maximizing statistical power. To evaluate the power of the proposed DCT-GMM, we first generate $X$ and $Y$ as independent pairs following a normal distribution, with mean and variance randomly sampled from a uniform distribution $U(0, 1)$. We then generate the conditioning variable $\mathbf{Z}$ as $Z_i = a_i X + b_i Y + E_i$, where $a_i$ and $b_i$ are scalars randomly drawn from a standard normal distribution, and $E_i$ follows a standard normal distribution, i.e., $X \not\!\perp\!\!\!\perp Y \mid \mathbf{Z}$. The same discretization approach is applied here. The last two columns of Figure 2 illustrate the Type II error rates for both varying sample sizes and changing cardinalities of the conditioning set scenarios.

According to the first row of Figure 2, DCT-GMM (for both steps) and DCT show superior performance in maintaining Type I error close to the significance level across all sample sizes and conditioning set cardinalities, while other baselines, which do not account for discretization, exhibit significantly higher Type I errors. As the sample size in-

creases, both the Chi-Square and Fisher-Z tests tend to yield larger Type I errors because these methods measure whether $\tilde{X} \perp\!\!\!\perp \tilde{Y} \mid \tilde{\mathbf{Z}}$. More samples only reinforce incorrect conclusions. Additionally, DCT-GMM demonstrates significantly higher power compared to DCT, particularly for small sample sizes. This advantage arises from DCT-GMM's ability to utilize more information by solving multiple equations, highlighting its superiority and effectiveness.

### 4.2. Application in Causal Discovery

CI testing plays a pivotal role in causal discovery, which aims to uncover causal relationships from observational data. Two fundamental assumptions—faithfulness and the causal Markov condition—allow causal structures, represented by a Directed Acyclic Graph (DAG) $\mathcal{G}$, to be inferred from statistical independence relations. Based on these principles, constraint-based methods like the PC algorithm (Spirtes et al., 2000) recover graph structures through CI testing. However, discretization compromises the reliability of CI tests, leading to incorrect dependence assertions and distorting the inferred DAG.

To validate the effectiveness of DCT-GMM, we follow the setting of (Sun et al., 2024) and apply the PC algorithm with different CI testing methods on a synthetic dataset. Specifically, the true DAG is generated using the Bipartite Pairing (BP) model (Asratian et al., 1998), with weights drawn from a uniform distribution $U \sim (1, 3)$ and incorporating noise following a standard normal distribution. The number of

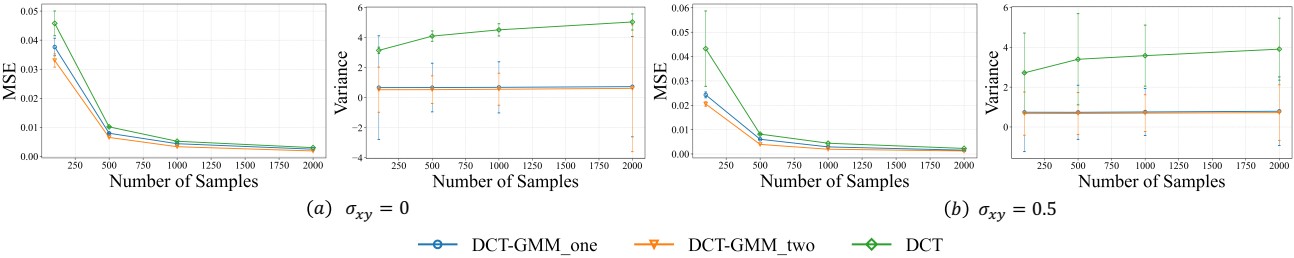

*Figure 4.* Comparison of Variance and MSE of estimated covariance using DCT and DCT-GMM.

edges in the DAG is one fewer than the number of nodes. While this graph is relatively sparse, the main focus of DCT-GMM is to correct CIs incorrectly judged as conditional dependence due to discretization. As the number of edges increases, such cases of true CI become rarer. We also investigate the performance of DCT-GMM in denser graphs, detailed in Appendix E.1.

The continuous data is then discretized into three levels, with boundaries randomly generated according to each variable's support. The experiment is divided into two cases: In the first, we fix the sample size at $n = 2000$ while varying the number of nodes $p = (4, 6, 8, 12)$. In the second, we fix the number of nodes at $p = 10$ and explore sample sizes of $n = (100, 500, 1000, 2000)$.

We compare DCT-GMM (both steps) against the Fisher-Z test (Fisher, 1921), the Chi-square test (F.R.S., 2009), and the previous work DCT (Sun et al., 2024) applied to discretized data. Additionally, we apply the Fisher-Z test to the original continuous data as a theoretical upper bound. Since the PC algorithm can only identify the causal graph up to a Completed Partially Directed Graph (CPDAG), we apply the same orientation rule from (Dor & Tarsi, 1992), implemented by (Chandler Squires, 2018), to convert the returned CPDAG to a DAG for easier comparison. For each setting, we run 10 graph instances with different seeds and report the mean and standard deviation of F1-score, precision, recall, and Structural Hamming Distance (SHD) in Figure 3 for skeleton discovery and Figure 5 for DAG comparison in Appendix D.

Experimental results show that DCT-GMM consistently outperforms DCT across all metrics, particularly recall, aligning with its higher statistical power demonstrated in Section 4.1. Moreover, DCT-GMM significantly outperforms the Chi-square and Fisher-Z tests, especially at large sample sizes, where the performance of these traditional tests deteriorates as the sample size increases. This phenomenon underscores the importance of discretization-aware CI test: *for all tests not aware of discretization, increasing the sample size only reinforces incorrect conclusions with greater confidence.* The lower recall observed in DCT-GMM and DCT compared to other baselines is expected, as other base-

lines tend to misinterpret conditional independence as dependence, leading to denser inferred graph structures.

### 4.3. Empirical Comparison of DCT and DCT-GMM

To validate the superiority of DCT-GMM over DCT, we conducted experiments to examine the variance and Mean Square Error (MSE) of their respective covariance estimators. Both methods adopt the same nodewise regression framework to transition from $\hat{\sigma}_{j_1 j_2} - \sigma^*_{j_1 j_2}$ to $\hat{\beta}_{j,k} - \beta^*_{j,k}$. Therefore, evaluating the covariance estimator directly reflects the performance of the CI test.

For a given pair of variables $X$ and $Y$, we denote the covariance estimated by DCT as $\hat{\sigma}^D_{XY}$ and that by DCT-GMM as $\hat{\sigma}^G_{XY}$. We empirically assess the stability and accuracy of these estimators by comparing their variance and MSE across varying sample sizes $n = (100, 200, 500, 1000, 2000)$ under the scenario that the true covariance $\sigma_{XY} = (0, 0.5)$ and fixed discretization level $M = 3$. The Figure 4 shows the results.

As shown in Figure 4, DCT-GMM consistently outperforms DCT. Specifically, the variance of $\hat{\sigma}^G_{XY}$ is consistently lower than that of $\hat{\sigma}^D_{XY}$ across all evaluated sample sizes. Additionally, the MSE of $\hat{\sigma}^G_{XY}$ is smaller, indicating a more accurate estimation. These experimental results validate the superiority and improved sample efficiency of DCT-GMM, providing support for Theorem 3.5.

## 5. Conclusion

In this paper, we propose DCT-GMM, a novel sample-efficient CI test to address challenges in CI testing with discretized data. By formulating parameter estimation as an overidentification problem and leveraging GMM, DCT-GMM surpasses existing methods, achieving lower estimation variance and greater statistical power, particularly in small-sample scenarios. Its proven effectiveness in causal discovery highlights its practical utility, bridging the gap between discretized observations and latent variable relationships in both synthetic and real-world datasets.

## Impact Statement

This paper presents work whose goal is to advance the field of Machine Learning. There are many potential societal consequences of our work, none which we feel must be specifically highlighted here.

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

*Appendix for*

# "A Sample-Efficient Conditional Independence Test in the Presence of Discretization"

## A. Notation Table and Function Form

### A.1. Notation Table

| Category | Description |
|---|---|
| **Number and Indices** | |
| $n$ | Number of samples |
| $p$ | Number of variables |
| $j_1, j_2, j, k$ | Index of a variable $j_1, j_2, j, k \in (1, \ldots, p)$ |
| **Random Variables** | |
| $\boldsymbol{X}$ | A vector of Gaussian variables |
| $\tilde{\boldsymbol{X}}$ | The discretized counterparts of $\boldsymbol{X}$ |
| $\boldsymbol{\Sigma}$ | Covariance of $\boldsymbol{X}$ |
| $\boldsymbol{\Sigma}_{-j-j}$ | Submatrix of $\boldsymbol{\Sigma}$ with $j$-th row and $j$-th column removed |
| $\boldsymbol{\Sigma}_{-jj}$ | $j$-th column of $\boldsymbol{\Sigma}$ with $j$-th row removed |
| $\boldsymbol{\Omega}$ | Precision matrix of $\boldsymbol{X}$, equals to $\boldsymbol{\Sigma}^{-1}$ |
| $\boldsymbol{X}_{-\{jk\}}$ | All other variables of $\boldsymbol{X}$ with $X_j$ and $X_k$ removed |
| $\mathbf{c}_j$ | The discretization boundaries mapping $X_j$ to $\tilde{X}_j$ |
| $X_j$ | $j$-th component of the $\boldsymbol{X}$ |
| $\sigma_{j_1 j_2}$ | Covariance between $X_{j_1}$ and $X_{j_2}$ |
| $\omega_{jk}$ | Precision coefficient $\omega_{jk}$ |
| $x_j^i$ | $i$-th sample of $X_j$ |
| $\tilde{x}_j^i$ | $i$-th sample of $\tilde{X}_j$ |
| $c_{j,k}$ | One component of $\mathbf{c}_j$ |
| $\tau_{j,k}$ | True probability of $\tilde{X}_j$ has the value $k$ |
| $\tau_{j_1 j_2, mk}$ | True probability of $\tilde{X}_{j_1}$ has the value $m$, and $\tilde{X}_{j_2}$ has the value $k$. |
| $\beta_{j,k}$ | Regression coefficient of $X_k$ in predicting $X_j$ |
| $\beta_j$ | Vector of all coefficients regressing $X_j$ |
| $\xi_{j_1 j_2}^i$ | Influence function component, it represents the influence of the $i$-th observation on the covariance estimation error |
| $\boldsymbol{\Xi}^i$ | Matrix form of $\xi^i$ |
| **Estimation of Variables** | |
| $\hat{\sigma}_{j_1 j_2}$ | Estimated covariance of $X_{j_1}, X_{j_2}$ |
| $\hat{\boldsymbol{\Sigma}}$ | Estimated covariance matrix, also the matrix form of $\hat{\sigma}_{j_1 j_2}$ |
| $\hat{\omega}_{jk}$ | Estimation of $\omega_{jk}$ |
| $\hat{c}_{j,k}$ | Estimation of $c_{j,k}$, calculated using Equation (2) |
| $\hat{\tau}_{j,k}$ | Estimation of $\tau_{j,k}$, the sample probability that $\tilde{X}_j$ equals to $k$ |
| $\hat{\tau}_{j_1 j_2, mk}$ | Estimation of $\tau_{j_1 j_2, mk}$, the sample probability that $\tilde{X}_{j_1}$ equals to $m$ and $\tilde{X}_{j_2}$ equals to $k$ |
| $\hat{\beta}_j$ | Estimation of $\hat{\beta}_j$, calculated as $\hat{\boldsymbol{\Sigma}}_{-j-j}^{-1} \hat{\boldsymbol{\Sigma}}_{-jj}$ |
| **Functions and Operators** | |
| $\mathbb{P}$ | True probability |
| $\mathbb{P}_n$ | Sample probability |
| $\mathbb{E}[Z]$ | Expectation of a random variable $Z$ |
| $\mathbb{E}_n[Z]$ | Sample mean of a random variable $Z$ over $n$ samples |
| $\mathbb{1}$ | Indicator function: is 1 if the condition is true, 0 otherwise |
| $\Phi(z)$ | Cumulative distribution function of a standard normal distribution integrated from $-\infty$ to $z$ |

| Category | Description |
| --- | --- |
| $\Phi(a, b, c, d; \sigma_{j_1 j_2})$ | Cumulative distribution function of a bivariate normal distribution with covariance $\sigma_{j_1 j_2}$ integrated over the rectangular region defined by $[a, b] \times [c, d]$. |
| **Notations of GMM** | |
| $f_i(\boldsymbol{\theta})$ | Moment function defined in 4 |
| $g_i(\boldsymbol{\theta})$ | Sample mean of moment functions given $n$ samples $g_i(\boldsymbol{\theta}) = \frac{1}{n} \sum_{i=1}^{n} f_i(\boldsymbol{\theta})$ |
| **A** | Weighting matrix of GMM |
| **G** | The expectation of the Jacobian of the moment function $\mathbf{G} = \mathbb{E}[\frac{\partial f_i(\boldsymbol{\theta}^*)}{\partial \boldsymbol{\theta}}]$ |
| **S** | Covariance matrix of moment function $\mathbf{S} = \mathbb{E}[f_i(\boldsymbol{\theta}^*)f_i(\boldsymbol{\theta}^*)^T]$ |

### A.2. Cumulative Distribution Function of Bivariate Normal Distribution

The probability density function of a bivariate normal distribution with random variables $X_{j_1}, X_{j_2}$, mean $\mathbf{0}$, unit variance and covariance $\sigma_{j_1 j_2}$ is given by:

$$\phi(x_{j_1}, x_{j_2}; \sigma_{j_1 j_2}) = \frac{1}{\sqrt{1 - \sigma_{j_1 j_2}^2}} \exp\left(-\frac{x_{j_1}^2 - 2\sigma_{j_1 j_2} x_{j_1 j_2} + x_{j_2}^2}{2(1 - \sigma_{j_1 j_2}^2)}\right). \tag{12}$$

The cdf of the bivariate normal distribution with the covariance $\sigma_{j_1 j_2}^*$ integrating over the rectangular region defined by $[c_{j_1, m-1}^*, c_{j_1, m}^*] \times [c_{j_2, k-1}^*, c_{j_2, k}^*]$ is

$$\Phi(c_{j_1, m-1}^*, c_{j_1, m}^*, c_{j_2, k-1}^*, c_{j_2, k}^*; \sigma_{j_1 j_2}^*) = \int_{c_{j_1, m-1}^*}^{c_{j_1, m}^*} \int_{c_{j_2, k-1}^*}^{c_{j_2, k}^*} \phi(x_{j_1}, x_{j_2}; \sigma_{j_1 j_2}^*) dx_{j_1} dx_{j_2}. \tag{13}$$

# B. Discussion of Assumptions

**Rationality of the assumption**    A primary limitation of this work lies in the assumption of a multivariate normal distribution for the latent continuous variables, which, to some extent, restricts its generality. However, it is important to emphasize the inherent challenge of proposing a valid conditional independence test in a discretization scenario without relying on appropriate assumptions.

To accurately infer conditional independence relationships within this framework, three key components are required: 1. A meaningful statistic — capable of capturing the conditional independence among latent variables. 2. A consistent estimator — the statistic must be computable solely from discretized observations. 3. Statistical inference — the null distribution of the statistic must be derivable. The discretization drastically reduces available information, making these components less straightforward to implement compared to scenarios where all variables are directly observable.

Without a parametric assumption, deriving a meaningful statistic is already challenging, let alone performing its statistical inference. We adopt the same framework of (Sun et al., 2024), relying on the property that with a parametric assumption, the covariance of the original latent variables is computable, and for Gaussian variables, the covariance matrix corresponds to the independence and conditional independence among variables.

**w.l.o.g of the assumption**    One question that may intrigue readers is why the assumption of zero mean and unit variance is made without loss of generality. The answer is straightforward: we can always adjust the discretization function and its boundaries to produce equivalent results for models with non-zero means and varying variances.

To illustrate, consider an intuitive example. Suppose we observe a discrete variable $\tilde{X}_j$ with $n$ samples, where half are labeled as "ones" and the other half as "twos." This discrete distribution could correspond to multiple continuous variables. For instance, a continuous variable $X_j \sim N(0, 1)$ with a discretization boundary at 0 and another variable $X'_j \sim N(1, 2)$ with a discretization boundary at 1 would yield exactly the same discretized observations $\tilde{X}_j$. Thus, the framework presented in this paper supports mapping the same discretized observations to multivariate normal distributions with any mean and variance, i.e., the zero mean and unit variance assumption is without loss of generality.

## C. Pseudo Code

---

**Algorithm 1** one step DCT-GMM

---

1: **Require:**

- Observed data matrix $\tilde{\boldsymbol{X}}' \in \mathbb{R}^{n \times d}$

- Tested pair indices $j_1, j_2$ with $j_1 \neq j_2$

- Conditioning set $\mathbf{C} \subseteq \{1, \ldots, d\} \setminus \{j_1, j_2\}$

- Significance level $\alpha$

2: **Rearrange Data Matrix**
$$\tilde{\boldsymbol{X}} = \left[ \tilde{\boldsymbol{X}}'[:, j_1], \tilde{\boldsymbol{X}}'[:, j_2], \tilde{\boldsymbol{X}}'[:, \mathbf{C}] \right] \in \mathbb{R}^{n \times p}, \quad \text{where } p = 2 + |\mathbf{C}|$$

3: **Initialize Covariance Matrix**
$$\hat{\boldsymbol{\Sigma}} \leftarrow \mathbf{I}_p \quad \text{(identity matrix of size } p \times p)$$

4: **for** $q \leftarrow 1$ **to** $p$ **do**
5:     **for** $k \leftarrow q + 1$ **to** $p$ **do**
6:         Obtain the Cardinality of $\boldsymbol{X}'[:, q]$ as $Q$
7:         Obtain the Cardinality of $\boldsymbol{X}'[:, k]$ as $K$
8:         Set the naive weighting matrix $A \leftarrow \mathbf{I}_{QK}$
9:         Compute covariance $\hat{\sigma}_{qk}$ through minimizing Equation (5)
10:        Update covariance matrix:
$$\hat{\boldsymbol{\Sigma}}[q, k] \leftarrow \hat{\sigma}_{qk} \qquad \hat{\boldsymbol{\Sigma}}[k, q] \leftarrow \hat{\sigma}_{qk} \quad \text{(ensuring symmetry)}$$

11:     **end for**
12: **end for**
13: **Extract Submatrices ($j_1$ and $j_2$ correspond the first and second column of $\tilde{X}$ due to the regroup)**

- Let $\hat{\boldsymbol{\Sigma}}_{-1-1} \in \mathbb{R}^{p-1 \times p-1} \leftarrow$ the submatrix of $\hat{\boldsymbol{\Sigma}}$ without 1st column and 1st row

- Let $\hat{\boldsymbol{\Sigma}}_{-11} \in \mathbb{R}^{p-1}$ be the 1st column of $\hat{\boldsymbol{\Sigma}}$ with first row removed

14: **Compute Test Statistics**
$$\hat{\beta}_{1,2} \leftarrow \hat{\boldsymbol{\Sigma}}_{-1-1}^{-1} \hat{\boldsymbol{\Sigma}}_{-11}$$

15: **Formulate Null Distribution**

$$\Phi(z) \leftarrow \text{Cumulative distribution function of the Normal Distribution defined in Thm. 3.4}$$

16: **Calculate p-value**
$$p\text{-value} \leftarrow 2 \cdot \left( 1 - \Phi\left( |\hat{\beta}_{1,2}| \right) \right)$$

17: **Make Decision**
18: **if** $p$-value $> \alpha$ **then**
19:     **Conclude**: $X_{j_1} \perp\!\!\!\perp X_{j_2} \mid X_{\mathbf{S}}$
20: **else**
21:     **Conclude**: $X_{j_1} \not\!\perp\!\!\!\perp X_{j_2} \mid X_{\mathbf{S}}$
22: **end if**
23: **Return** The conditional independence decision

---

---

**Algorithm 2** two step DCT-GMM

---

1: **Require:**

- Observed data matrix $\tilde{\boldsymbol{X}}' \in \mathbb{R}^{n \times d}$

- Tested pair indices $j_1, j_2$ with $j_1 \neq j_2$

- Conditioning set $\mathbf{C} \subseteq \{1, \ldots, d\} \setminus \{j_1, j_2\}$

- Significance level $\alpha$

2: **Rearrange Data Matrix**
$$\tilde{\boldsymbol{X}} = \left[ \tilde{\boldsymbol{X}}'[:, j_1], \tilde{\boldsymbol{X}}'[:, j_2], \tilde{\boldsymbol{X}}'[:, \mathbf{C}] \right] \in \mathbb{R}^{n \times p}, \quad \text{where } p = 2 + |\mathbf{C}|$$

3: **Initialize Covariance Matrix**
$$\hat{\boldsymbol{\Sigma}} \leftarrow \mathbf{I}_p \quad \text{(identity matrix of size } p \times p\text{)}$$

4: **for** $q \leftarrow 1$ **to** $p$ **do**
5:    **for** $k \leftarrow q + 1$ **to** $p$ **do**
6:       Obtain the Cardinality of $\boldsymbol{X}'[:, q]$ as $Q$
7:       Obtain the Cardinality of $\boldsymbol{X}'[:, k]$ as $K$
8:       Set the naive weighting matrix $A \leftarrow \mathbf{I}_{QK}$
9:       Obtain estimated parameters $\hat{\boldsymbol{\theta}}$ through minimizing Equation (5)
10:      Set the weighting matrix $A \leftarrow \mathbb{E}_n[f_i(\hat{\boldsymbol{\theta}}) f_i(\hat{\boldsymbol{\theta}})^T]$, where $f_i(\boldsymbol{\theta})$ defined in Equation (4)
11:      Resolve Equation (5) with updated $A$ to calculate the estimated covariance $\hat{\sigma}_{qk}$
12:      Update covariance matrix:
$$\hat{\boldsymbol{\Sigma}}[q, k] \leftarrow \hat{\sigma}_{qk} \qquad \hat{\boldsymbol{\Sigma}}[k, q] \leftarrow \hat{\sigma}_{qk} \quad \text{(ensuring symmetry)}$$

13:    **end for**
14: **end for**
15: **Extract Submatrices ($j_1$ and $j_2$ correspond the first and second column of $\tilde{X}$ due to the regroup)**

- Let $\hat{\boldsymbol{\Sigma}}_{-1-1} \in \mathbb{R}^{p-1 \times p-1} \leftarrow$ the submatrix of $\hat{\boldsymbol{\Sigma}}$ without 1st column and 1st row

- Let $\hat{\boldsymbol{\Sigma}}_{-11} \in \mathbb{R}^{p-1}$ be the 1st column of $\hat{\boldsymbol{\Sigma}}$ with first row removed

16: **Compute Test Statistics**
$$\hat{\beta}_{1,2} \leftarrow \hat{\boldsymbol{\Sigma}}_{-1-1}^{-1} \hat{\boldsymbol{\Sigma}}_{-11}$$

17: **Formulate Null Distribution**
$$\Phi(z) \leftarrow \text{Cumulative distribution function of the Normal Distribution defined in Thm. 3.4}$$

18: **Calculate p-value**
$$p\text{-value} \leftarrow 2 \cdot \left( 1 - \Phi\left( |\hat{\beta}_{1,2}| \right) \right)$$

19: **Make Decision**
20: **if** $p$-value $> \alpha$ **then**
21:    **Conclude**: $X_{j_1} \perp\!\!\!\perp X_{j_2} \mid X_{\mathbf{S}}$
22: **else**
23:    **Conclude**: $X_{j_1} \not\perp\!\!\!\perp X_{j_2} \mid X_{\mathbf{S}}$
24: **end if**
25: **Return** The conditional independence decision

---

# D. Figure of Main Experiments: Causal Discovery

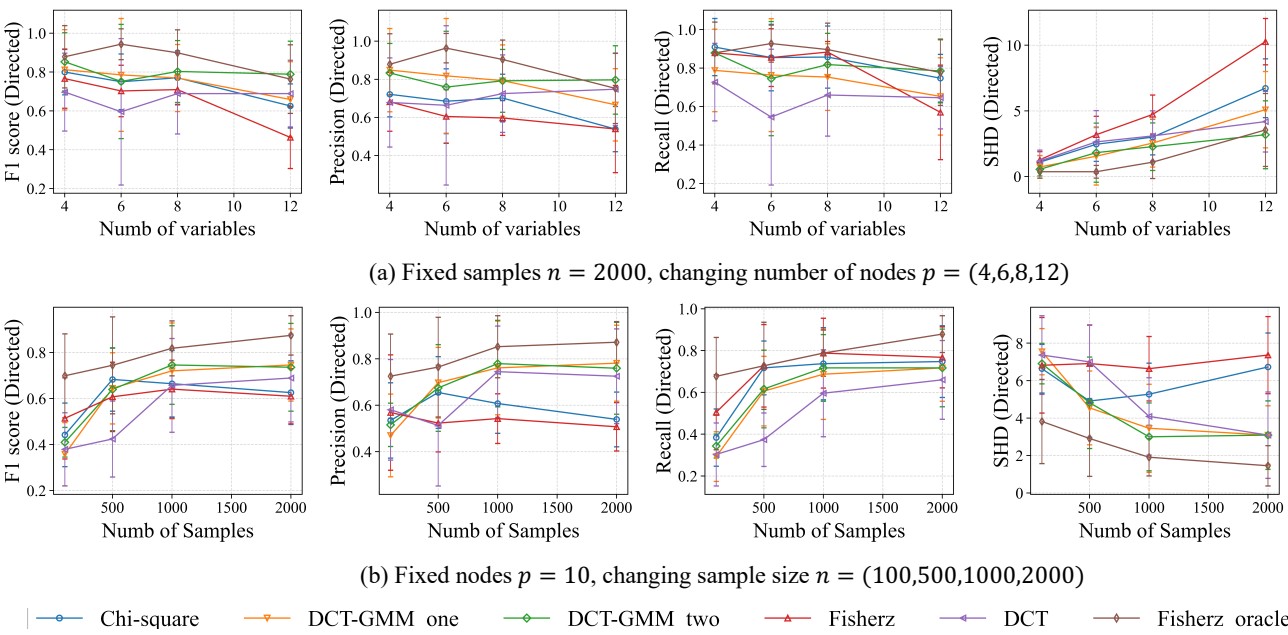

(a) Fixed samples $n = 2000$, changing number of nodes $p = (4,6,8,12)$

(b) Fixed nodes $p = 10$, changing sample size $n = (100,500,1000,2000)$

Chi-square    DCT-GMM one    DCT-GMM two    Fisherz    DCT    Fisherz oracle

*Figure 5.* Experimental result of DAG discovery on synthetic data for changing number of nodes (a) and changing sample size(b). Fisherz_oracle is the Fisher-z test applied to original continuous data. We evaluate $F_1$ (↑), Precision (↑), Recall (↑) and SHD (↓).

# E. Additional Experiments

## E.1. Denser Graph

DCT-GMM is most effective in cases where discretization causes true conditional independencies to be incorrectly identified as dependencies. Its performance is therefore particularly strong in sparse graph settings, where true conditional independence relationships are abundant. However, to comprehensively evaluate a test's statistical power—its ability to correctly identify true conditional dependencies—it is crucial to examine its performance in dense graph scenarios. To this end, we conduct experiments with $p = 10$ nodes and $n = 2000$ samples, varying the edge density $(p+2, p+4, p+6, p+8)$. The underlying continuous data follows a multivariate Gaussian distribution, with the true DAG $\mathcal{G}$ generated using the BP model. We perform 10 independent trials with different random seeds and present both skeleton discovery and DAG reconstruction results in Fig. 6.

Experimental results show that DCT-GMM continues to outperform other baselines in terms of precision and SHD. As the number of edges increases, the advantage of discretization-aware CI tests (DCT-GMM and DCT) gradually diminishes due to the decreasing prevalence of conditional independence cases. Notably, DCT-GMM maintains superior recall, consistent with the findings from the main causal discovery experiment.

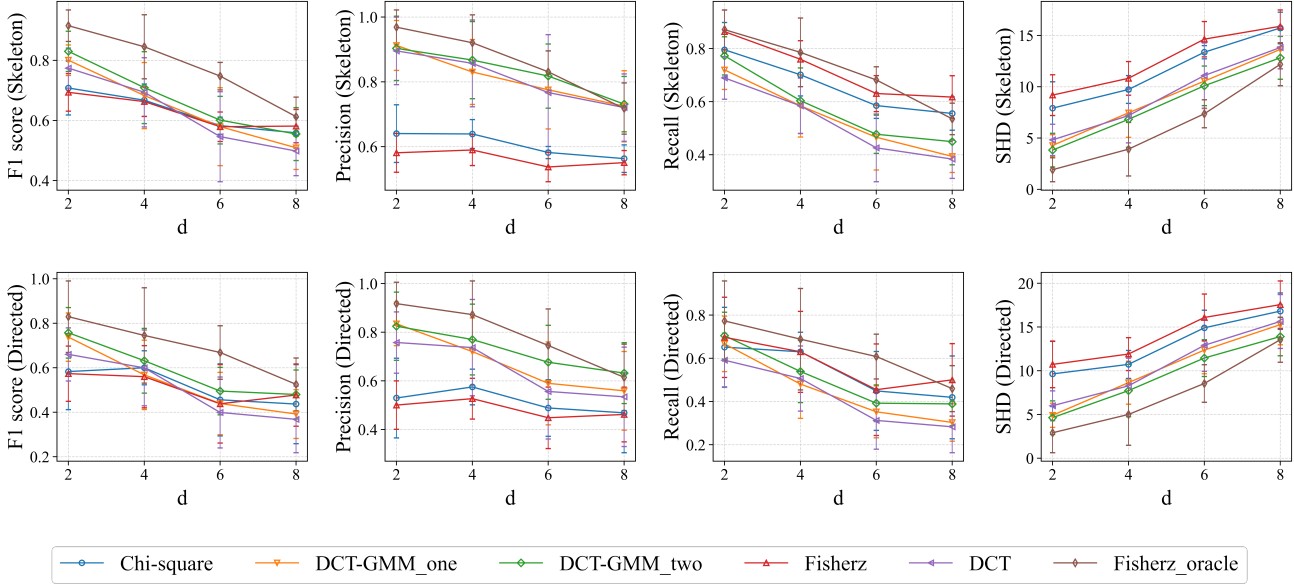

*Figure 6.* Experimental comparison of causal discovery on synthetic datasets for denser graphs with $p = 10, n = 2000$ and edges varying $p+2, p+4, p+6, p+8$. We evaluate F1 ($\uparrow$), Precision ($\uparrow$), Recall ($\uparrow$) and SHD ($\downarrow$) on both skeleton and DAG.

## E.2. Real-world Experiment

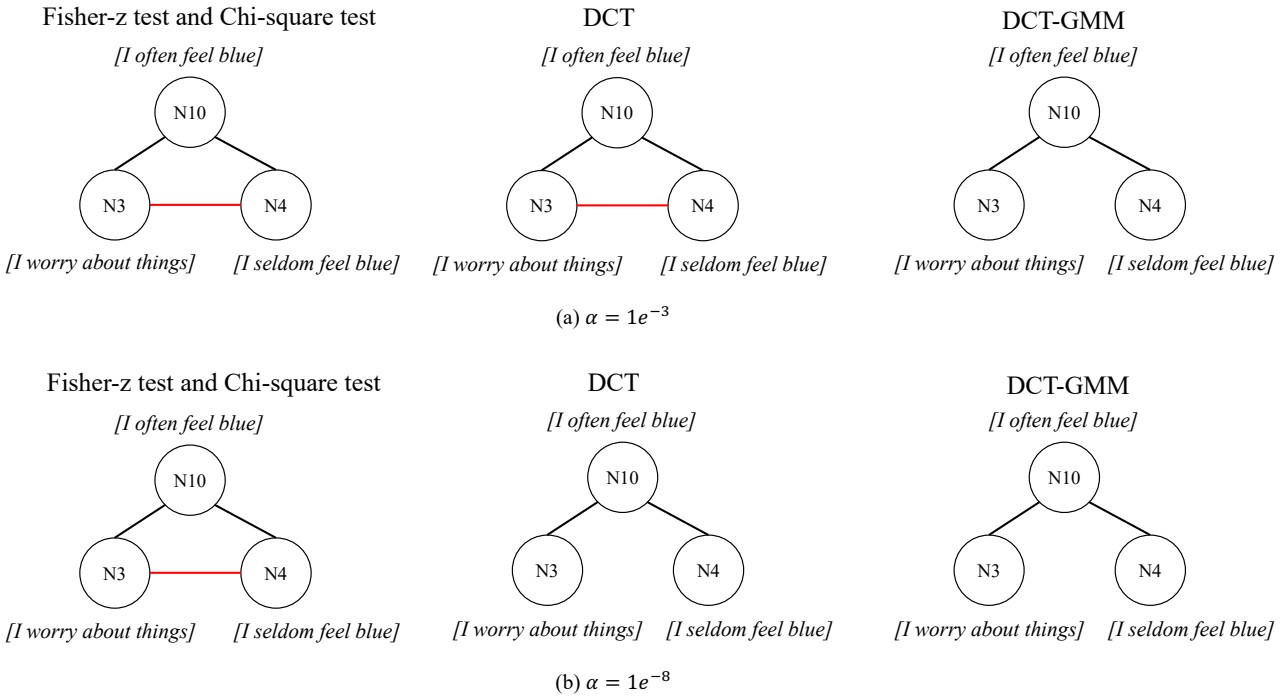

*Figure 7.* PC algorithm applied on the real-world dataset with Fisher-z test, Chi-square test, DCT and DCT-GMM for different significance level $\alpha$. Red edge are found by other baselines while DCT-GMM removes.

To validate the effectiveness of DCT-GMM, we conduct experiments on the Big Five Personality dataset, where each variable has 5 discrete values representing agreement levels (1=Disagree to 5=Agree). For example, "N3=1" indicates "I disagree that I worry about things.". This setting aligns well with DCT-GMM, as agreement levels are inherently continuous but observed as discrete categories. This dataset has been closely examined by Dong et al. (2024a) and Dong et al. (2024b), yet it does not solve the discretization problem. We focus on three variables: [N3: I worry about things], [N10: I often feel blue], and [N4: I seldom feel blue]. Using the PC algorithm for causal discovery, we compare DCT-GMM with the Chi-square and Fisher-Z tests. Results are shown in Fig. 7.

Experimental results validate both the effectiveness and superiority of DCT-GMM. Notably, both discretization-aware CI tests (DCT and DCT-GMM) successfully remove the edge between $N3$ and $N4$, whereas other baselines fail. The inferred graph directly aligns with our motivating causal graph illustrated in Figure 1. Furthermore, DCT-GMM demonstrates a stronger ability to capture conditional independence relationships. Increasing the significance level $\alpha$ generally makes CI tests more prone to inferring conditional dependence. While DCT fails at $\alpha = 10^{-3}$, DCT-GMM remains robust, correctly identifying that $N3 \perp\!\!\!\perp N4 \mid N10$.

## F. Proof and Derivations

### F.1. Proof of Moment Condition

In this part, We show the derivation that $\mathbb{E}[f_i(\boldsymbol{\theta}^*)] = \mathbf{0}$. For the moment functions $f_i(\boldsymbol{\theta}^*)$ defined in Equation (4) with the parameters achieving their optimal $\boldsymbol{\theta} = \boldsymbol{\theta}^*$, we have the specific form:

$$
f_i(\boldsymbol{\theta}) = \begin{pmatrix} \hat{\tau}^i_{j_1 j_2, 11} - \Phi(c^*_{j_1,0}, c^*_{j_1,1}, c^*_{j_2,0}, c^*_{j_2,1}; \sigma^*_{j_1 j_2}) \\ \vdots \\ \hat{\tau}^i_{j_1 j_2, MK} - \Phi(c^*_{j_1,M-1}, c^*_{j_1,M}, c^*_{j_2,K-1}, c^*_{j_2,K}; \sigma^*_{j_1 j_2}) \end{pmatrix}.
$$

For any $m \in (1, \ldots, M), k \in (1, \ldots, K)$, the cdf term $\Phi(c^*_{j_1,m-1}, c^*_{j_1,m}, c^*_{j_2,k-1}, c^*_{j_2,k}; \sigma^*_{j_1 j_2})$ represents the area of this bivariate normal distribution integrated over the region defined by $[c^*_{j_1,m-1}, c^*_{j_1,m}] \times [c^*_{j_2,k-1}, c^*_{j_2,k}]$. In probability terms, this corresponds to

$$
\mathbb{P}(c^*_{j_1,m-1} < X_{j_1} < c^*_{j_1,m}, c^*_{j_2,k-1} < X_{j_2} < c^*_{j_2,k}).
$$

For its corresponding counterpart of the discrete domain, the relation holds

$$
\Phi(c^*_{j_1,m-1}, c^*_{j_1,m}, c^*_{j_2,k-1}, c^*_{j_2,k}; \sigma^*_{j_1 j_2}) = \mathbb{P}(\tilde{X}_{j_1} = m, \tilde{X}_{j_2} = k).
$$

Recall the definition that $\hat{\tau}^i_{j_1 j_2, mk} = \mathbb{1}(\tilde{x}^i_{j_1} = m, \tilde{x}^i_{j_2} = k)$ is the indicator function of the sample $i$. Its expectation is equivalent to the corresponding probability:

$$
\mathbb{E}[\hat{\tau}^i_{j_1 j_2, mk}] = \mathbb{P}(\tilde{X}_{j_1} = m, \tilde{X}_{j_2} = k).
$$

We note that $\Phi(\cdot; \sigma^*_{j_1 j_2})$ is a constant with respect to the sample, we can take expectations over $f_i(\boldsymbol{\theta}^*)$ term-wise:

$$
\mathbb{E}[f_i(\boldsymbol{\theta})] = \begin{pmatrix} \mathbb{E}[\hat{\tau}^i_{j_1 j_2, 11}] - \Phi(c^*_{j_1,0}, c^*_{j_1,1}, c^*_{j_2,0}, c^*_{j_2,1}; \sigma^*_{j_1 j_2}) \\ \vdots \\ \mathbb{E}[\hat{\tau}^i_{j_1 j_2, MK}] - \Phi(c^*_{j_1,M-1}, c^*_{j_1,M}, c^*_{j_2,K-1}, c^*_{j_2,K}; \sigma^*_{j_1 j_2}). \end{pmatrix}
$$

Substituting both $\mathbb{E}[\hat{\tau}^i_{j_1 j_2, 11}]$ and $\Phi(c^*_{j_1,0}, c^*_{j_1,1}, c^*_{j_2,0}, c^*_{j_2,1}; \sigma^*_{j_1 j_2})$ as $\mathbb{P}(\tilde{X}_{j_1} = m, \tilde{X}_{j_2} = k)$, each term evaluates to zero, giving:

$$
\mathbb{E}[f_i(\boldsymbol{\theta}^*)] = \mathbf{0}.
$$

This concludes the proof.

### F.2. Proof of Theorem 3.1

In this part, we show the detailed derivation of Theorem 3.1. Recall the definition of GMM defined in Equation (5), we are trying to minimize

$$
\hat{J}(\boldsymbol{\theta}) = \hat{g}(\boldsymbol{\theta})^T A \hat{g}(\boldsymbol{\theta}),
$$

where the $\hat{g}(\boldsymbol{\theta}) \in \mathbb{R}^{MK}$ is the sample mean of the moment functions, and the $\boldsymbol{\theta} \in \mathbb{R}^{M+K-1}$ is the interested parameters. When the interested parameter $\boldsymbol{\theta} = \hat{\boldsymbol{\theta}}$, we define the Jacobian matrix

$$
\hat{\mathbf{G}} = \frac{\partial \hat{g}(\hat{\boldsymbol{\theta}})}{\partial \hat{\boldsymbol{\theta}}} \in \mathbb{R}^{MK \times (M+K-1)}.
$$

Using the chain rule, we have

$$
\frac{\partial \hat{J}(\hat{\boldsymbol{\theta}})}{\partial \hat{\boldsymbol{\theta}}} = 2\hat{\mathbf{G}}^T \mathbf{A} \hat{g}(\hat{\boldsymbol{\theta}}).
$$

We note that when the interested parameter $\boldsymbol{\theta} = \hat{\boldsymbol{\theta}}$, which is the minimum of $\hat{J}(\boldsymbol{\theta})$, its gradient should be zero:

$$
2\hat{\mathbf{G}}^T \mathbf{A} \hat{g}(\boldsymbol{\theta}) = \mathbf{0}. \tag{14}
$$

Leveraging Taylor expansion, we have

$$\hat{g}(\boldsymbol{\theta}^*) = \hat{g}(\hat{\boldsymbol{\theta}}) + \hat{\mathbf{G}}(\boldsymbol{\theta}^* - \hat{\boldsymbol{\theta}}) + \dots,$$

where the second-order terms and more are omitted. Rearrange the equation above, we have

$$\hat{g}(\hat{\boldsymbol{\theta}}) = \hat{g}(\boldsymbol{\theta}^*) - \hat{\mathbf{G}}(\boldsymbol{\theta}^* - \hat{\boldsymbol{\theta}}). \tag{15}$$

Substituting back into the first-order condition Equation (14), the Equation (15) becomes

$$2\hat{\mathbf{G}}^T \mathbf{A} \left( \hat{g}(\boldsymbol{\theta}^*) - \hat{\mathbf{G}}(\boldsymbol{\theta}^* - \hat{\boldsymbol{\theta}}) \right) = \mathbf{0}. \tag{16}$$

Simplify and rearrange terms, we have the difference between the estimator and the true parameter:

$$\hat{\boldsymbol{\theta}} - \boldsymbol{\theta}^* = -(\hat{\mathbf{G}}^T \mathbf{A} \hat{\mathbf{G}})^{-1} \hat{\mathbf{G}}^T \mathbf{A} \hat{g}(\boldsymbol{\theta}^*)$$

$$= -\frac{1}{n} \sum_{i=1}^{n} (\hat{\mathbf{G}}^T \mathbf{A} \hat{\mathbf{G}})^{-1} \hat{\mathbf{G}}^T \mathbf{A} f_i(\boldsymbol{\theta}^*).$$

According to the Central Limit Theorem, when $n \to +\infty$, the sample average of the moment functions should be asymptotically normal:

$$\frac{1}{n} \sum_{i=1}^{n} f_i(\boldsymbol{\theta}^*) \xrightarrow{d} N(\mathbf{0}, \mathbb{E}[f_i(\boldsymbol{\theta}^*) f_i(\boldsymbol{\theta}^*)^T]/n).$$

Since the mean is zero due to the definition. We note that the Jacobian term

$$\hat{\mathbf{G}} = \frac{\partial \hat{g}(\hat{\boldsymbol{\theta}})}{\partial \hat{\boldsymbol{\theta}}} = \mathbb{E}_n[\frac{\partial f_i(\hat{\boldsymbol{\theta}})}{\partial \hat{\boldsymbol{\theta}}}]$$

due to the sample terms are irrelevant with the parameter $\boldsymbol{\theta}$. According to the Law of large numbers, when $n \to +\infty$, the estimated parameter $\hat{\boldsymbol{\theta}} \xrightarrow{p} \boldsymbol{\theta}^*$. Thus, the Jacobian

$$\hat{\mathbf{G}} \xrightarrow{p} \mathbf{G} := \mathbb{E}[\frac{\partial f_i(\boldsymbol{\theta}^*)}{\partial \boldsymbol{\theta}^*}].$$

Let $\mathbf{S} := \mathbb{E}[f_i(\boldsymbol{\theta}^*) f_i(\boldsymbol{\theta}^*)^T]$ for simplicity of the notation, according to Slutsky's theorem, we have

$$\sqrt{n}(\hat{\boldsymbol{\theta}} - \boldsymbol{\theta}^*) \xrightarrow{d} N\left(\mathbf{0}, (\mathbf{G}^T \mathbf{A} \mathbf{G})^{-1} \mathbf{G}^T \mathbf{A} \mathbf{S} \mathbf{A} \mathbf{G} (\mathbf{G}^T \mathbf{A} \mathbf{G})^{-1}\right). \tag{17}$$

Since $\sigma_{j_1 j_2}$ is nothing but the first element of the $\boldsymbol{\theta}$, we conclude that

$$\sqrt{n}(\hat{\sigma}_{j_1 j_2} - \sigma_{j_1 j_2}) \xrightarrow{d} N\left(0, \left[(\mathbf{G}^T \mathbf{A} \mathbf{G})^{-1} \mathbf{G}^T \mathbf{A} \mathbf{S} \mathbf{A} \mathbf{G} (\mathbf{G}^T \mathbf{A} \mathbf{G})^{-1}\right]_{11}\right), \tag{18}$$

which concludes the proof.

### F.3. Proof of Lemma 3.2

In this part, we show the detailed derivation of Lemma 3.2. Our proof is divided in to two parts: we first show the specific form of variance will follow when $\mathbf{A} \xrightarrow{p} \mathbf{S}^{-1}$. We then establish it superiority by showing $(\mathbf{G}^T \mathbf{A} \mathbf{G})^{-1} \mathbf{G}^T \mathbf{A} \mathbf{S} \mathbf{A} \mathbf{G} (\mathbf{G}^T \mathbf{A} \mathbf{G})^{-1} \succeq (\mathbf{G}^T \mathbf{S}^{-1} \mathbf{G})^{-1}$.

When $n \to +\infty$, the positive semi-definite weighting matrix $\mathbf{A}$ converges to the $\mathbf{S}^{-1}$, the variance of the original asymptotical defined in Theorem 3.1, will be written as:

$$\begin{aligned}
(\mathbf{G}^T \mathbf{A} \mathbf{G})^{-1} \mathbf{G}^T \mathbf{A} \mathbf{S} \mathbf{A} \mathbf{G} (\mathbf{G}^T \mathbf{A} \mathbf{G})^{-1} &= (\mathbf{G}^T \mathbf{S}^{-1} \mathbf{G})^{-1} \mathbf{G}^T \mathbf{S}^{-1} \mathbf{S} \mathbf{S}^{-1} \mathbf{G} (\mathbf{G}^T \mathbf{S}^{-1} \mathbf{G})^{-1} \\
&= (\mathbf{G}^T \mathbf{S}^{-1} \mathbf{G})^{-1} \mathbf{G}^T \mathbf{S}^{-1} \mathbf{G} (\mathbf{G}^T \mathbf{S}^{-1} \mathbf{G})^{-1} \\
&= (\mathbf{G}^T \mathbf{S}^{-1} \mathbf{G})^{-1}.
\end{aligned} \tag{19}$$

That is,

$$\sqrt{n}(\hat{\boldsymbol{\theta}} - \boldsymbol{\theta}^*) \xrightarrow{d} N\left(\mathbf{0}, (\mathbf{G}^T\mathbf{S}^{-1}\mathbf{G})^{-1}\right). \tag{20}$$

Since $\sigma_{j_1 j_2}$ is nothing but the first element of the $\boldsymbol{\theta}$, we conclude that

$$\sqrt{n}(\hat{\sigma}_{j_1 j_2} - \sigma_{j_1 j_2}) \xrightarrow{d} N\left(0, \left[(\mathbf{G}^T\mathbf{S}^{-1}\mathbf{G})^{-1}\right]_{11}\right), \tag{21}$$

which concludes the first part of the proof. We now dive into the second part.

First, we factor $\mathbf{S} = \mathbf{C}\mathbf{C}^T$, where $\mathbf{C} \in \mathbb{R}^{MK \times MK}$ which is non-singular. Second, we let

$$\mathbf{H} = (\mathbf{G}^T\mathbf{A}\mathbf{G})^{-1}\mathbf{G}\mathbf{C} - (\mathbf{G}^T\mathbf{S}^{-1}\mathbf{G})^{-1}\mathbf{G}^T\mathbf{C}^{-T}.$$

Third, we not that

$$\mathbf{H}\mathbf{C}^{-1}\mathbf{G} = \mathbf{0}.$$

Fourth, we verify that

$$(\mathbf{G}^T\mathbf{A}\mathbf{G})^{-1}\mathbf{G}^T\mathbf{A}\mathbf{S}\mathbf{A}\mathbf{G}(\mathbf{G}^T\mathbf{A}\mathbf{G})^{-1} = \mathbf{H}\mathbf{H}^T + (\mathbf{G}^T\mathbf{S}^{-1}\mathbf{G})^{-1}.$$

Since $\mathbf{H}\mathbf{H}^T$ is positive semi-definite, the $(\mathbf{G}^T\mathbf{S}^{-1}\mathbf{G})^{-1}$ is a lower bound of $(\mathbf{G}^T\mathbf{A}\mathbf{G})^{-1}\mathbf{G}^T\mathbf{A}\mathbf{S}\mathbf{A}\mathbf{G}(\mathbf{G}^T\mathbf{A}\mathbf{G})^{-1}$, which concludes the proof.

### F.4. Proof of Thm. 3.4

We note that the following proof is a direct copy from (Sun et al., 2024). We include it here for completeness.

#### F.4.1. DERIVATION OF EQUATION 8

Consider our latent continuous variables $\boldsymbol{X} = (X_1, \ldots, X_p) \sim N(0, \boldsymbol{\Sigma})$ and do nodewise regression

$$X_j = \boldsymbol{X}_{-j}\beta_j + \epsilon_j, \tag{22}$$

where $\boldsymbol{X}_{-j}$ is the submatrix of $\boldsymbol{X}$ with $X_j$ removed. We can divide its covariance $\boldsymbol{\Sigma}$ and its precision matrix $\Omega = \boldsymbol{\Sigma}^{-1}$ into the predictor $\mathbf{X}_{-j}$ and outcome variable $X_j$ in our regression:

$$\boldsymbol{\Sigma} = \begin{pmatrix} \boldsymbol{\Sigma}_{jj} & \boldsymbol{\Sigma}_{j-j} \\ \boldsymbol{\Sigma}_{-jj} & \boldsymbol{\Sigma}_{-j-j} \end{pmatrix} \quad \boldsymbol{\Omega} = \begin{pmatrix} \boldsymbol{\Omega}_{jj} & \boldsymbol{\Omega}_{j-j} \\ \boldsymbol{\Omega}_{-jj} & \boldsymbol{\Omega}_{-j-j} \end{pmatrix}. \tag{23}$$

Just like regular linear regression, we can get

$$n \to \infty, \quad \beta_j = \boldsymbol{\Sigma}_{-j-j}^{-1}\boldsymbol{\Sigma}_{-jj}. \tag{24}$$

From the invertibility of a block matrix

$$\begin{bmatrix} A & B \\ C & D \end{bmatrix}^{-1} = \begin{bmatrix} (A - BD^{-1}C)^{-1} & -(A - BD^{-1}C)^{-1}BD^{-1} \\ -D^{-1}C(A - BD^{-1}C)^{-1} & D^{-1} + D^{-1}C(A - BD^{-1}C)^{-1}BD^{-1} \end{bmatrix}. \tag{25}$$

If $A$ and $D$ is invertible, we will have

$$\begin{bmatrix} A & B \\ C & D \end{bmatrix}^{-1} = \begin{bmatrix} (A - BD^{-1}C)^{-1} & 0 \\ 0 & (D - CA^{-1}B)^{-1} \end{bmatrix} \begin{bmatrix} I & -BD^{-1} \\ -CA^{-1} & I \end{bmatrix}. \tag{26}$$

Thus, we can get:

$$\boldsymbol{\Omega}_{jj} = (\boldsymbol{\Sigma}_{jj} - \boldsymbol{\Sigma}_{j-j}\boldsymbol{\Sigma}_{-j-j}^{-1}\boldsymbol{\Sigma}_{-jj})^{-1};$$
$$\boldsymbol{\Omega}_{j-j} = -\left(\boldsymbol{\Sigma}_{jj} - \boldsymbol{\Sigma}_{j-j}\boldsymbol{\Sigma}_{-j-j}^{-1}\boldsymbol{\Sigma}_{-jj}\right)^{-1}\boldsymbol{\Sigma}_{j-j}(\boldsymbol{\Sigma}_{-j-j})^{-1}. \tag{27}$$

Move one step forward:

$$-\mathbf{\Omega}_{jj}^{-1}\mathbf{\Omega}_{j-j} = \mathbf{\Sigma}_{j-j}(\mathbf{\Sigma}_{-j-j})^{-1}. \tag{28}$$

Take transpose for both sides, as long as $\mathbf{\Omega}$ is a symmetric matrix and $\mathbf{\Omega}_{-jj} = \mathbf{\Omega}_{j-j}^{T}$, we will have

$$-\mathbf{\Omega}_{jj}^{-1}\mathbf{\Omega}_{-jj} = \mathbf{\Sigma}_{-j-j}^{-1}\mathbf{\Sigma}_{-jj} = \beta_j. \tag{29}$$

We should note testing $\mathbf{\Omega}_{-jj} = \mathbf{0}$ is equivalent to testing $\beta_j = \mathbf{0}$ as the $\mathbf{\Omega}_{jj}$ will always be nonzero. The variable $\mathbf{\Omega}_{-jj}$ captures the CI of $X_j$ with other variables. As long as the variable $\mathbf{\Omega}_{jj}$ is just one scalar, we can get

$$\beta_{j,k} = -\frac{\omega_{jk}}{\omega_{jj}} \tag{30}$$

capturing the CI relationship between variable $X_j$ with $X_k$ conditioning on all other variables.

### F.4.2. DETAILED DERIVATION OF INFERENCE FOR $\beta_j$

Nodewise regression allows us to use the regression parameter $\beta_j$ as the surrogate of $\mathbf{\Omega}_{-jj}$. The problem now transfers to constructing the inference for $\beta_j$, specifically, the derivation of distribution of $\hat{\beta}_j - \beta_j$. The overarching concept is that we are already aware of the distribution of $\hat{\sigma}_{j_1 j_2} - \sigma_{j_1 j_2}$ and we know that there exists a deterministic relationship between $\beta_j$ with $\mathbf{\Sigma}$. Consequently, we can express $\hat{\beta}_j - \beta_j$ as a composite of $\hat{\sigma}_{j_1 j_2} - \sigma_{j_1 j_2}$ to establish such an inference. Specifically, we have

$$\begin{aligned}
\hat{\beta}_j - \beta_j &= \hat{\mathbf{\Sigma}}_{-j-j}^{-1}\hat{\mathbf{\Sigma}}_{-jj} - \mathbf{\Sigma}_{-j-j}^{-1}\mathbf{\Sigma}_{-jj} \\
&= \hat{\mathbf{\Sigma}}_{-j-j}^{-1}\left(\hat{\mathbf{\Sigma}}_{-jj} - \hat{\mathbf{\Sigma}}_{-j-j}\mathbf{\Sigma}_{-j-j}^{-1}\mathbf{\Sigma}_{-jj}\right) \\
&= -\hat{\mathbf{\Sigma}}_{-j-j}^{-1}\left(\hat{\mathbf{\Sigma}}_{-j-j}\beta_j - \mathbf{\Sigma}_{-j-j}\beta_j + \mathbf{\Sigma}_{-j-j}\beta_j - \hat{\mathbf{\Sigma}}_{-jj}\right) \\
&= -\hat{\mathbf{\Sigma}}_{-j-j}^{-1}\left((\hat{\mathbf{\Sigma}}_{-j-j} - \mathbf{\Sigma}_{-j-j})\beta_j - (\hat{\mathbf{\Sigma}}_{-jj} - \mathbf{\Sigma}_{-jj})\right),
\end{aligned} \tag{31}$$

where each entry in matrix $(\hat{\mathbf{\Sigma}}_{-j-j} - \mathbf{\Sigma}_{-j-j})$ and $(\hat{\mathbf{\Sigma}}_{-jj} - \mathbf{\Sigma}_{-jj})$ denotes the difference between estimated covariance with true covariance.

Suppose that we want to test the CI of the variable $X_1$ with other variables, $j = 1$. then

$$\hat{\mathbf{\Sigma}}_{-1-1} - \mathbf{\Sigma}_{-1-1} = \begin{bmatrix} \hat{\sigma}_{22} \dots \hat{\sigma}_{2p} \\ \dots \\ \hat{\sigma}_{p2} \dots \hat{\sigma}_{pp} \end{bmatrix} - \begin{bmatrix} \sigma_{22} \dots \sigma_{2p} \\ \dots \\ \sigma_{p2} \dots \sigma_{pp} \end{bmatrix} \tag{32}$$

$$:= \frac{1}{n}\sum_{i=1}^{n} \begin{bmatrix} \xi_{22}^{i} \dots \xi_{2p}^{i} \\ \dots \\ \xi_{p2}^{i} \dots \xi_{pp}^{i} \end{bmatrix}, \tag{33}$$

where $\{\xi_{j_1 j_2}^i\}$ are i.i.d random variables with specific form defined in Theorem 3.1 for one-step GMM and Lemma 3.2 for two-step GMM correspondingly. Put them together:

$$\hat{\beta}_1 - \beta_1 = \begin{bmatrix} \hat{\beta}_{1,2} - \beta_{1,2} \\ \hat{\beta}_{1,3} - \beta_{1,3} \\ \dots \\ \hat{\beta}_{1,p} - \beta_{1,p} \end{bmatrix} = -\hat{\mathbf{\Sigma}}_{-1-1}^{-1}\frac{1}{n}\sum_{i=1}^{n}\left(\begin{bmatrix} \xi_{22}^{i} & \xi_{23}^{i} & \cdots & \xi_{2,p}^{i} \\ \xi_{32}^{i} & \xi_{33}^{i} & \cdots & \xi_{3p}^{i} \\ \dots & \dots & \dots & \dots \\ \xi_{p2}^{i} & \xi_{p3}^{i} & \cdots & \xi_{pp}^{i} \end{bmatrix}\begin{bmatrix} \beta_{1,2} \\ \beta_{1,3} \\ \dots \\ \beta_{1,p} \end{bmatrix} - \begin{bmatrix} \xi_{21}^{i} \\ \xi_{31}^{i} \\ \dots \\ \xi_{p1}^{i} \end{bmatrix}\right). \tag{34}$$

As $\frac{1}{n}\sum_{i=1}^{n}\xi_{j_1 j_2}^{i}$ is asymptotically normal, the who vector of $\hat{\beta}_1 - \beta_1$ is a linear combination of Gaussian distribution. However, We cannot merely engage in a linear combination of its variance as they are dependent with each other. For example, if $Y_1, Y_2$ are dependent and we are trying to find out $Var(aY_1 + bY_2)$, we should have

$$Var(aY_1 + bY_2) = \begin{bmatrix} a & b \end{bmatrix}\begin{bmatrix} Var(Y_1) & Cov(Y_1, Y_2) \\ Cov(Y_1, Y_2) & Var(Y_2) \end{bmatrix}\begin{bmatrix} a \\ b \end{bmatrix}. \tag{35}$$

Now, suppose we are interested in the distribution of $\hat{\beta}_{1,2} - \beta_{1,2}$, we have

$$\hat{\beta}_{1,2} - \beta_{1,2} = \frac{1}{n}\sum_{i=1}^{n}(\hat{\mathbf{\Sigma}}_{-1-1}^{-1})_{[2],:}\left(\begin{bmatrix} \xi_{2,2}^i & \xi_{2,3}^i & \cdots & \xi_{2,p}^i \\ \xi_{3,2}^i & \xi_{3,3}^i & \cdots & \xi_{3,p}^i \\ \cdots & \cdots & \cdots & \cdots \\ \xi_{p,2}^i & \xi_{p,3}^i & \cdots & \xi_{p,p}^i \end{bmatrix}\begin{bmatrix} \beta_{1,2} \\ \beta_{1,3} \\ \cdots \\ \beta_{1,p} \end{bmatrix} - \begin{bmatrix} \xi_{2,1}^i \\ \xi_{3,1}^i \\ \cdots \\ \xi_{p,1}^i \end{bmatrix}\right), \tag{36}$$

where $(\hat{\mathbf{\Sigma}}_{-1-1}^{-1})_{[2],:}$ is the row of index of $X_2$ of $\hat{\mathbf{\Sigma}}_{-1-1}^{-1}$ ([2] denotes the index of the variable, e.g., $(\hat{\mathbf{\Sigma}}_{-1,-1}^{-1})_{[2],:}$ represents the first row of $\hat{\mathbf{\Sigma}}_{-1,-1}^{-1}$ since the row of first variable is removed. ). For ease of notation, we define

$$\mathbf{Y}^i = \mathbf{\Xi}_{-1,-1}^i = \begin{bmatrix} \xi_{2,2}^i & \xi_{2,3}^i & \cdots & \xi_{2,p}^i \\ \xi_{3,2}^i & \xi_{3,3}^i & \cdots & \xi_{3,p}^i \\ \cdots & \cdots & \cdots & \cdots \\ \xi_{p,2}^i & \xi_{p,3}^i & \cdots & \xi_{p,p}^i \end{bmatrix} \in \mathbb{R}^{p-1 \times p-1}, \qquad \mathbf{v}^i := \mathbf{\Xi}_{-1,1}^i = \begin{bmatrix} \xi_{2,1}^i \\ \xi_{3,1}^i \\ \cdots \\ \xi_{p,1}^i \end{bmatrix} \in \mathbb{R}^{p-1}, \tag{37}$$

$$\mathbf{u} := (\hat{\mathbf{\Sigma}}_{-1,-1}^{-1})_{[2],:}^T \in \mathbb{R}^{p-1} \qquad \mathbf{w} := \begin{bmatrix} \beta_{1,2} \\ \beta_{1,3} \\ \cdots \\ \beta_{1,p} \end{bmatrix} \in \mathbb{R}^{p-1}.$$

We can rewrite the equation as

$$\hat{\beta}_{1,2} - \beta_{1,2} = -\frac{1}{n}\sum_{l=1}^{n}\mathbf{u}(\mathbf{Y}^i\mathbf{w} - \mathbf{v}^i).$$

We note that $\mathbf{Y}^i$, $\mathbf{v}^i$ are variables, and $\mathbf{u}$, $\mathbf{w}$ are constants (just like the example $aY_1 + bY_2$). We further let $m = p - 1$ to simplify the notation. We can thus write the equation above as vector form:

$$\hat{\beta}_{1,2} - \beta_{1,2} = -\frac{1}{n}\sum_{l=1}^{n}\left[u_1, \ldots, u_m, u_1w_1, u_1w_2, \ldots, u_mw_m\right]\begin{bmatrix} -v_1^i \\ \cdots, \\ -v_m^i \\ Y_{11}^i \\ Y_{12}^i \\ \cdots \\ Y_{mm}^i \end{bmatrix}$$

$$= -\frac{1}{n}\sum_{i=1}^{n}[\mathbf{u}^T, \text{vec}(\mathbf{uw}^T)^T]\begin{bmatrix} -\mathbf{v}^i \\ \text{vec}(\mathbf{Y}^i)) \end{bmatrix},$$

where $u_k$ represents the $k$-th element of vector $\mathbf{u}$ and $Y_{jk}^i$ represents the entry in $j$-th row and $k$-th column of matrix $\mathbf{Y}^i$, vec represents the row-wise vectorization of a matrix, e.g,

$$\text{vec}(\mathbf{Y}^l) = \begin{bmatrix} Y_{11} \\ Y_{12} \\ Y_{13} \\ \cdots \\ Y_{mm} \end{bmatrix} \in \mathbb{R}^{m^2}.$$

Similar as equation 35, the variance is calculated as

$$Var\left(\sqrt{n}(\hat{\beta}_{1,2} - \beta_{1,2})\right) = \frac{1}{n}\sum_{l=1}^{n}[\mathbf{u}^T, \text{vec}(\mathbf{uw}^T)^T]\begin{bmatrix} -\mathbf{v}^l \\ \text{vec}(\mathbf{Y}^i) \end{bmatrix}\begin{bmatrix} -\mathbf{v}^l \\ \text{vec}(\mathbf{Y}^i) \end{bmatrix}^T\begin{bmatrix} \mathbf{u} \\ \text{vec}(\mathbf{uw}^T) \end{bmatrix}.$$

Now we go back to use the notations of $\xi$ and $\mathbf{\Sigma}$. Under the null hypothesis that $X_1 \perp\!\!\!\perp X_2 | X_{others}$, i.e., $\beta_{1,2} = 0$. We thus use $\tilde{\boldsymbol{\beta}}_1$ to denote $\boldsymbol{\beta}_1$ where $\beta_{1,2} = 0$. Let

$$\mathbf{B}_{-1}^i = \begin{pmatrix} \xi_{21}^i & \xi_{31}^i & \cdots & \xi_{p1}^i \\ \xi_{22}^i & \xi_{23}^i & \cdots & \xi_{2p}^i \\ \xi_{32}^i & \xi_{33}^i & \cdots & \xi_{3p}^i \\ \cdots & \cdots & \cdots & \cdots \\ \xi_{p2}^i & \xi_{p3}^l & \cdots & \xi_{pp}^i \end{pmatrix} = \begin{bmatrix} \mathbf{\Xi}_{-11}^i{}^T \\ \mathbf{\Xi}_{-1-1}^i \end{bmatrix},$$

and

$$\boldsymbol{a}^{[2]} = \begin{bmatrix} -(\hat{\boldsymbol{\Sigma}}^{-1}_{-1,-1})^T_{[2],:} \\ \text{vec}\left( (\hat{\boldsymbol{\Sigma}}^{-1}_{-1,-1})^T_{[2],:} \tilde{\boldsymbol{\beta}}^T_1 \right) \end{bmatrix}$$

Similarly as (35), The variance is calculated as

$$Var\left( \sqrt{n}(\hat{\beta}_{1,2} - \beta_{1,2}) \right) = \boldsymbol{a}^{[2]T} \frac{1}{n} \sum_{l=1}^{n} \text{vec}(\boldsymbol{B}^i_{-1}) \text{vec}(\boldsymbol{B}^i_{-1})^T \boldsymbol{a}^{[2]},$$

Simply replace the index $1, 2$ as general index $j, k$, the distribution of $\hat{\beta}_{j,k} - \beta_{j,k}$ is

$$\hat{\beta}_{j,k} - \beta_{j,k} \xrightarrow{d} N(0, \boldsymbol{a}^{[k]T} \frac{1}{n^2} \sum_{l=1}^{n} \text{vec}(\boldsymbol{B}^i_{-j}) \text{vec}(\boldsymbol{B}^i_{-j})^T) \boldsymbol{a}^{[k]}).$$

In practice, we can plug in the estimates of $\beta_j$ to estimate the interested distribution and do the CI test by hypothesizing $\beta_{j,k} = 0$.

# G. Formal Claim of Theorem 3.5 and Derivation

In this section, we try to demonstrate the theoretical advantage of DCT-GMM over DCT. Specifically, the variance of $\hat{\beta}_{j,k} - \beta_{j,k}^*$ obtained using DCT-GMM is consistently lower than that of DCT. Since DCT-GMM and DCT adopt exactly the same strategy to transition from $\hat{\sigma}_{j_1 j_2} - \sigma_{j_1 j_2}^*$ to $\hat{\beta}_{j,k} - \beta_{j,k}^*$, which is simply a linear combination of $\hat{\sigma}_{j_1 j_2} - \sigma_{j_1 j_2}^*$. A reduction in the variance of $\hat{\sigma}_{j_1 j_2} - \sigma_{j_1 j_2}^*$ directly translates to a reduction in the variance of $\hat{\beta}_{j,k} - \beta_{j,k}^*$ is. Thus, our task is to prove the variance of the estimator of covariance using DCT, denoted $\text{Var}_{\text{DCT}}(\hat{\sigma}_{j_1 j_2} - \sigma_{j_1 j_2}^*)$, is consistently greater than the one of two-step DCT-GMM, denoted $\text{Var}_{\text{GMM}}(\hat{\sigma}_{j_1 j_2} - \sigma_{j_1 j_2}^*)$.

The proof is organized into two parts:

1. **Review of Variance Derivation of DCT**: We first provide an review of the derivation for $\text{Var}_{\text{DCT}}(\hat{\sigma}_{j_1 j_2} - \sigma_{j_1 j_2}^*)$. 2. **Moment Function Selection**: Next, we show that with appropriate moment functions, $\text{Var}_{\text{DCT}}(\hat{\sigma}_{j_1 j_2} - \sigma_{j_1 j_2}^*)$ equals $\text{Var}_{\text{GMM}}(\hat{\sigma}_{j_1 j_2} - \sigma_{j_1 j_2}^*)$. We then directly use the property of GMM that incorporating valid moment functions lead to less variance, which concludes the proof.

## G.1. Review of Variance Derivation of DCT

We begin with the derivation of $\text{Var}_{\text{DCT}}(\hat{\sigma}_{j_1 j_2} - \sigma_{j_1 j_2}^*)$ with a particular focus on discrete case. For discretized observed variable pair $\tilde{X}_{j_1}$ and $\tilde{X}_{j_2}$, DCT implicitly treats it as a pair of binary variables. Recall the definitions in DCT, we have interested parameters $\boldsymbol{\theta} = (\sigma_{j_1 j_2}, h_{j_1}, h_{j_2})$, with the function

$$g(\boldsymbol{\theta}) = \frac{1}{n} \sum_{i=1}^{n} f_i(\boldsymbol{\theta}) = \begin{pmatrix} \hat{\tau}_{j_1 j_2}^i - \bar{\Phi}(h_{j_1}, h_{j_2}; \sigma_{j_1 j_2}) \\ \hat{\tau}_{j_1}^i - \bar{\Phi}(h_{j_1}) \\ \hat{\tau}_{j_2}^i - \bar{\Phi}(h_{j_2}) \end{pmatrix}. \tag{38}$$

For the true parameters $\boldsymbol{\theta}^* = (\sigma_{j_1 j_2}^*, h_{j_1}^*, h_{j_2}^*)$, we have

$$g(\boldsymbol{\theta}^*) = \frac{1}{n} \sum_{i=1}^{n} f_i(\boldsymbol{\theta}^*) = \begin{pmatrix} \hat{\tau}_{j_1 j_2}^i - \bar{\Phi}(h_{j_1}^*, h_{j_2}^*; \sigma_{j_1 j_2}^*) \\ \hat{\tau}_{j_1}^i - \bar{\Phi}(h_{j_1}^*) \\ \hat{\tau}_{j_2}^i - \bar{\Phi}(h_{j_2}^*) \end{pmatrix}, \tag{39}$$

and the function of estimated parameters

$$g(\hat{\boldsymbol{\theta}}) = \frac{1}{n} \sum_{i=1}^{n} f_i(\hat{\boldsymbol{\theta}}) = \begin{pmatrix} \hat{\tau}_{j_1 j_2}^i - \bar{\Phi}(\hat{h}_{j_1}, \hat{h}_{j_2}; \hat{\sigma}_{j_1 j_2}) \\ \hat{\tau}_{j_1}^i - \bar{\Phi}(\hat{h}_{j_1}) \\ \hat{\tau}_{j_2}^i - \bar{\Phi}(\hat{h}_{j_2}) \end{pmatrix} = \mathbf{0}, \tag{40}$$

where

- $\hat{\tau}_{j_1 j_2}^i = \mathbb{1}(\tilde{x}_{j_1}^i > \mathbb{E}_n[\tilde{X}_{j_1}], \tilde{x}_{j_2}^i > \mathbb{E}_n[\tilde{X}_{j_2}])$ serving as the estimation of $\tau_{j_1 j_2}^i = \mathbb{1}(\tilde{x}_{j_1}^i > \mathbb{E}[\tilde{X}_{j_1}], \tilde{x}_{j_2}^i > \mathbb{E}[\tilde{X}_{j_2}])$

- $\hat{\tau}_{j_1}^i = \mathbb{1}(\tilde{x}_{j_1}^i > \mathbb{E}_n[\tilde{X}_{j_1}])$ serving as the estimation of $\tau_{j_1}^i = \mathbb{1}(\tilde{x}_{j_1}^i > \mathbb{E}[\tilde{X}_{j_1}])$.

- $\Phi(x, y; z) = \int_{-\infty}^{x} \int_{-\infty}^{y} \frac{1}{2\pi\sqrt{1-z^2}} (\exp -\frac{u_1^2 - 2zu_1 u_2 + u_2^2}{2(1-z^2)}) du_1 du_2$ is the cumulative distribution function of a bivariate normal distribution.

- $\bar{\Phi}(x) = 1 - \Phi(x)$, $\bar{\Phi}(x, y; z) = 1 - \Phi(x, y; z)$

- $\Phi(x) = \int_{-\infty}^{x} \frac{1}{\sqrt{2\pi}} \exp -\frac{u^2}{2} du$ is the cumulative distribution function of a standard normal distribution.

Our objective is to construct the distribution of $\hat{\sigma}_{j_1 j_2} - \sigma_{j_1 j_2}$, equivalently $\hat{\boldsymbol{\theta}} - \boldsymbol{\theta}$. By leveraging the Taylor expansion, we can construct the following equation

$$g(\hat{\boldsymbol{\theta}}) = g(\boldsymbol{\theta}^*) + \frac{\partial g(\boldsymbol{\theta}^*)}{\partial \boldsymbol{\theta}^*}(\hat{\boldsymbol{\theta}} - \boldsymbol{\theta}^*) + \dots \tag{41}$$

where $\frac{\partial g(\boldsymbol{\theta}^*)}{\partial \boldsymbol{\theta}^*}$ is the Jacobian matrix of function $g$ at $\boldsymbol{\theta}^*$. The second terms and more are omitted here. Rearrange terms, since $g(\hat{\boldsymbol{\theta}})$ equals to zero, we have

$$\hat{\boldsymbol{\theta}} - \boldsymbol{\theta}^* = \frac{\partial g(\boldsymbol{\theta}^*)}{\partial \boldsymbol{\theta}^*}^{-1} g(\boldsymbol{\theta}^*), \tag{42}$$

if the Jacobian is invertible, which will always be true in this framework. Express $g(\boldsymbol{\theta}^*)$ in vector form, we have

$$\hat{\boldsymbol{\theta}} - \boldsymbol{\theta}^* = \frac{\partial g(\boldsymbol{\theta}^*)}{\partial \boldsymbol{\theta}^*}^{-1} \frac{1}{n} \sum_{i=1}^{n} \begin{pmatrix} \hat{\tau}_{j_1 j_2}^i - \bar{\Phi}(h_{j_1}^*, h_{j_2}^*; \sigma_{j_1 j_2}^*) \\ \hat{\tau}_{j_1}^i - \bar{\Phi}(h_{j_1}^*) \\ \hat{\tau}_{j_2}^i - \bar{\Phi}(h_{j_2}^*) \end{pmatrix}. \tag{43}$$

When $n \to +\infty$, the first term Jacobian matrix $\frac{\partial g(\boldsymbol{\theta}^*)}{\partial \boldsymbol{\theta}^*}$ will converge to $E[\frac{\partial f_i(\boldsymbol{\theta}^*)}{\partial \boldsymbol{\theta}^*}]$. It's noteworthy that $E[f_i(\boldsymbol{\theta}^*)] = \mathbf{0}$ according to the definition. By leveraging the Central limit theorem, we have

$$n \to +\infty, \quad \frac{1}{n} \sum_{i=1}^{n} f_i(\boldsymbol{\theta}^*) \sim N\left(\mathbf{0}, \frac{1}{n} E[f_i(\boldsymbol{\theta}^*) f_i(\boldsymbol{\theta}^*)^T]\right). \tag{44}$$

Thus, we have

$$\hat{\boldsymbol{\theta}} - \boldsymbol{\theta}^* \sim N\left(\mathbf{0}, \frac{1}{n} E[\frac{\partial f_i(\boldsymbol{\theta}^*)}{\partial \boldsymbol{\theta}^*}]^{-1} E[f_i(\boldsymbol{\theta}^*) f_i(\boldsymbol{\theta}^*)^T] E[\frac{\partial f_i(\boldsymbol{\theta}^*)}{\partial \boldsymbol{\theta}^*}]^{-T}\right) \tag{45}$$

### G.2. Moment Function Selection and Additional Moment Functions

We note that this derivation process is pretty similar to the one using GMM. Intuitively, if the moment functions of GMM are the same as Equation (38), we may have a similar distribution. We now provide the formal statement of the Theorem 3.5:

**Theorem G.1.** *For GMM whose a subset of moment functions $g(\boldsymbol{\theta})$ are the same as Equation (38), with additional moment functions defined in Equation 4, have strictly less variance than DCT, whose variance is given in (45).*

We now provide the proof of the theorem above. With appropriate moment functions, the variance of the $\hat{\sigma}_{j_1 j_2} - \sigma_{j_1 j_2}$ gotten using two-step DCT-GMM, is exactly the same as $Var_{GMM}(\hat{\sigma}_{j_1 j_2} - \sigma_{j_1 j_2})$. Specifically, for the interested parameters $\boldsymbol{\theta} = (\sigma_{j_1 j_2}, h_{j_1}, h_{j_2})$, we define the moment function the same as Equation (38). Specifically, we are solving the following minimization problem:

$$\hat{\boldsymbol{\theta}} = \arg\min_{\boldsymbol{\theta}} \quad g(\boldsymbol{\theta})^T \mathbf{A} g(\boldsymbol{\theta}), \tag{46}$$

with the moment condition $E[f_i(\boldsymbol{\theta}^*)] = \mathbf{0}$ satisfied for the true parameters $\boldsymbol{\theta}^*$. According to Lemma 3.2,

$$\hat{\boldsymbol{\theta}} - \boldsymbol{\theta}^* \sim (\mathbf{0}, \frac{1}{n}(\mathbf{G}^T \mathbf{A} \mathbf{G})^{-1}) \tag{47}$$

for the two-step estimation where

- $\mathbf{G} = E[\frac{\partial f_i(\boldsymbol{\theta}^*)}{\partial \boldsymbol{\theta}^*}]$
- $\mathbf{S} = E[f_i(\boldsymbol{\theta}^*) f_i(\boldsymbol{\theta}^*)^T]$
- $\mathbf{A} = \mathbf{S}^{-1}$.

Since $\mathbf{G}$ is invertible [1], we can rewrite

$$(\mathbf{G}^T \mathbf{A} \mathbf{G})^{-1} = \mathbf{G}^{-1} \mathbf{S} \mathbf{G}^{-T}, \tag{48}$$

which is the same as the variance in Equation (45). That is, $Var_{GMM}(\hat{\sigma}_{j_1 j_2} - \sigma_{j_1 j_2}) = Var_{DCT}(\hat{\sigma}_{j_1 j_2} - \sigma_{j_1 j_2})$. However, GMM accommodates additional moment functions (solvable equations as in Equation 3). Based on the property of GMM (Newey, 2007), adding valid moment functions (moment functions defined in Equation 4) generally reduces the variance of the parameter estimates, which concludes the proof.

---

[1]One may check DCT for the analytic form of $\mathbf{G}$, which will be a triangular matrix with non-zero diagonal entries.

