# OpenReview forum: "A Sample Efficient Conditional Independence Test in the Presence of Discretization"
_ICML.cc/2025/Conference — ICML 2025 poster_

### Official Review · Reviewer_9WLL · 2025-03-09

**Overall Recommendation:** 4

**Summary:**

This paper addresses the critical challenge of conducting conditional independence tests on discretized data, where traditional methods often fail due to information loss from binning. The authors propose DCT-GMM, a new method leveraging the Generalized Method of Moments to infer latent continuous variable relationships without binarization. Theoretical guarantees for asymptotic normality and reduced estimator variance are provided, and experiments demonstrate superior performance over existing methods in both Type I and Type II error rates and causal discovery tasks.

**Claims And Evidence:**

Yes.

**Essential References Not Discussed:**

No.

**Experimental Designs Or Analyses:**

Yes, I checked the experiments in Section 4.

**Methods And Evaluation Criteria:**

Yes.

**Other Comments Or Suggestions:**

None.

**Other Strengths And Weaknesses:**

Strengths: The integration of GMM to resolve over-identification in discretized data is novel and theoretically sound. The two-step GMM approach optimally weights moment conditions, enhancing efficiency.

Weaknesses: The method assumes latent variables follow a multivariate Gaussian distribution. This greatly limits its applicability to non-Gaussian settings.

**Questions For Authors:**

1. Given that each experimental configuration was replicated 2,000 times, Figure 2 suggests that the proposed method exhibits slight size inflation. What might be causing this phenomenon?

2. In all experiments, the data are discretized into three levels. How does the number of discretization levels impact the results?

**Relation To Broader Scientific Literature:**

The key contributions of the paper can advance methodologies in conditional independence testing, causal discovery, and handling discretized data.

**Theoretical Claims:**

No.

---

> ### Author Rebuttal · Authors · 2025-03-31
>
> > Multivariate Gaussian distribution.
>
> Thank you for raising the concern. We fully agree that the assumption of Gaussianity will limit the generality of the proposed test. At the same time, please allow us to share a few points regarding its reasonableness:
> 1. **Challenges in Conditional Independence in the Presence of Discretization**: Inferring the conditional independence of latent variables based on their discretized values is indeed challenging, despite being a common practical issue. _Discretization significantly reduces the available information._ Without introducing mild assumptions, it is particularly difficult—if not overly ambitious—to construct statistics that correctly reflect conditional independence of latent continuous variables, let alone develop valid inference procedures. In this work, we rely on two key properties:
> 	(1) The Gaussian assumption enables consistent estimation of the latent covariance matrix from discretized observations thanks to its parametric structure.
> 	(2) Under the Gaussian model, conditional independence can be inferred solely from the covariance matrix.
> 2.  **Popularity of nonparanormal Model**:  The assumption of latent variables following multivariate Gaussian, also called the nonparanormal model, is well-studied and widely accepted in the community. There is a substantial body of work demonstrating the effectiveness of the nonparanormal model in various scenarios [1,2].
> 3. **Empirical Performance:** To alleviate your concern and thanks to the insightul suggestion from Reviewer Nwpp, we conducted experiments investigating the Type I and Type II error of the proposed test where the data generation process violates our assumption. Specifically, the data are generated as either linear or nonlinear non-Gaussian, where the linear parameters follow the same setting of the main experiment and the nonlinear functions are randomly choose from $(a) f(x) = sin(x), (b) f(x) = x^3, (c) f(x) = tanh(x), (d) f(x) = ReLu(x)$.  Figure~1 in the link https://anonymous.4open.science/r/DCT-GMM-0D6D shows the comparison.
>     From the experiment result, DCT-GMM demonstrates comparable or superior Type I error control relative to DCT. In terms of Type II errors, it also outperforms DCT under most distributions. Overall, the discretization-aware tests clearly outperform other baselines.
>
> > Slight size inflation. What might be causing this phenomenon?
>
> We appreciate the reviewer for the valuable question. The most plausible explanation towards this phenomenon is the neglect of the second order and higher order terms in the deriving distribution of $ \hat{\sigma} - \sigma^*$ (Theorem 3.1 and Lemma 3.2 ). Specifically, our derivation relies on a Taylor expansion of the form (kindly refer to line 1045-1048):
> $$ \hat{g}(\mathbf{\theta}^*) = \hat{g}(\hat{\mathbf{\theta}}) + \hat{\mathbf{G}}(\mathbf{\theta}^* - \hat{\mathbf{\theta}}) + \dots,$$
> In our derivation, we  omit higher order terms, which  might influence the accuracy of the derived distribution. To validate the hypothesis, we conducted the experiment where we further increase the sample size, and strengthen the influence from the conditioning set to the observed pairs, thereby reducing the impact of higher-order terms.  As shown in Figure 4 (link: https://anonymous.4open.science/r/DCT-GMM-0D6D), the proposed test effectively controls the Type I error rate under these conditions, supporting the validity of our explanation.
> We will acknowledge this approximation as a limitation of our approach in the revised version of the paper.
>
> > How does the number of discretization levels impact the results?
>
> Thank you for the insightful question. Towards this question, we conducted additional experiments shown in Figure2 and Figure3 in the anonymous link https://anonymous.4open.science/r/DCT-GMM-0D6D.
>
> Figure 2 compares Type I and II errors of DCT-GMM and baselines, using discretization level $M=5$, the cardinality of conditioning set $D=1$ and change the sample size $n=(100,500,1000,2000)$. Similar to the main experiment, both DCT and DCT-GMM control Type I error well, while DCT-GMM achieves higher power.
>
> Figure 3 varies $M=(4,5,6,7,8)$ and fix $D=1$, and $n=2000$ in Figure3. DCT-GMM consistently maintains Type I error control and high power. Notably, the two-step DCT-GMM outperforms the one-step version, supporting our theoretical results.
>
> ---
>
> [1] Fan, J., Liu, H., Ning, Y., and Zou, H. High dimensional semiparametric latent graphical model for mixed data.   Journal of the Royal Statistical Society Series B: Statistical Methodology, 79(2):405–421, 2017.
>
> [2] Zhang A, Fang J, Hu W, et al. A latent Gaussian copula model for mixed data analysis in brain imaging genetics[J]. IEEE/ACM transactions on computational biology and bioinformatics, 2019, 18(4): 1350-1360.
>
> ---
>
> Thank you again for your thoughtful question. We hope the additional results clarify our findings. Please feel free to reach out if you have further questions or feedback.

---

> > ### Comment · Reviewer_9WLL · 2025-04-08
> >
> > Thanks for the response. I have raised my score.

---

### Official Review · Reviewer_bfoQ · 2025-03-11

**Overall Recommendation:** 3

**Summary:**

This paper introduces a Conditional Independence (CI) test designed for scenarios where continuous data is represented at a discretized level due to measurement limitations. In such settings, applying standard CI tests directly can lead to incorrect conclusions. Assuming the continuous data follows a multivariate normal distribution, the paper addresses this issue by proposing a Discretization-Aware CI test that accounts for these limitations and can identify independence relations between continuous latent variables given their discretized versions. This is achieved using the Generalized Method of Moments (GMM), which leverages all available sample information to estimate the covariance matrix of the continuous variables.

By assuming the original continuous variables follow a Gaussian model, performing the CI test reduces to an inference problem for the precision matrix $\Omega = \Sigma^{-1} = (w_{jk})$, since under this model, $w_{jk} = 0$ implies $X_j \perp\kern-0.3em\perp X_k \mid X_{-jk}$, where $X_{-jk}$ represents all other variables in $X$ except $X_j$ and $X_k$. The paper provides asymptotic error bounds on estimating these parameters and empirically demonstrates the improved performance of the proposed method compared to other CI tests.

**Claims And Evidence:**

yes, however, the multivariate normal distribution assumption should be stated more explicitly and earlier in the paper, as it is a key part of the analysis. It may be worth mentioning it in the abstract or contributions section since, as it stands, the language suggests a more general claim up until the assumption is introduced in line 130.

**Essential References Not Discussed:**

no

**Experimental Designs Or Analyses:**

Yes, results sounds reasonable

**Methods And Evaluation Criteria:**

yes

**Other Comments Or Suggestions:**

Minor comments:

- In Equation (1), it should be clarified that $m$ ranges from $2$ to $M-1$, or dots should be added in the brackets to indicate the full range. Initially, it appears as if there are only three cases instead of $M$ cases.

- In Lemma 3.3, Equation (8), the definitions for $\Sigma_{-j j}$ and $\Sigma^{-1}_{-j -j}$ should be stated explicitly in the lemma statement or at least referenced in the appendix (Equation 23). Currently, it is not immediately clear how these terms are defined until reaching Equation (23).

**Other Strengths And Weaknesses:**

Strengths:

-  Proposing a sample-efficient CI test that does not require binarization by utilizing GMM, making it more effective in preserving information from the original data.

- Promising empirical results



Weaknesses:

- The the multivariate normal distribution assumption limits the implication of the theoretical results
- From the technical perspective, the contribution is rather limited since the main results either uses standard asymptotic tools or results from (1)
- No guarantees for the type-I or type-II errors since the results are asymptotic and under the gaussian assumptions which may not hold in practice


(1)  Sun, B., Yao, Y., Hao, H., Qiu, Y., and Zhang, K. (2024). A conditional independence test in the presence of discretization.

**Questions For Authors:**

In the two-step procedure, is there a specific reason for limiting the process to only two steps, rather than iterating further until some form of convergence in the weight matrix is achieved? or the convergence itself is not guaranteed ?

**Relation To Broader Scientific Literature:**

The paper is related to the literature on Discretization-Aware CI tests (DCT), particularly the work in (1). However, the test in (1) relies on binarizing the observed data. The key contribution of this paper is proposing a sample-efficient CI test that does not require binarization, making it more effective in preserving information from the original data.

(1)  Sun, B., Yao, Y., Hao, H., Qiu, Y., and Zhang, K. (2024). A conditional independence test in the presence of discretization.

**Theoretical Claims:**

I went over the proofs but haven't checked line by line.

---

> ### Author Rebuttal · Authors · 2025-03-31
>
> > The multivariate normal distribution assumption.
>
> Thank you for raising the insightful concern. Please kindly refer to our response~1 to Reviewer 9WLL for a detailed explanation. Due to the 5000-character limit, we were unable to include the full response here. We apologize for the inconvenience.
>
> > The technical contribution is rather limited since the main results use standard asymptotic tools.
>
> Thank you for this important comment. We believe this comment arises from a lack of sufficient discussion in our paper. We have included additional discussion to clarify our use of asymptotic tools.
>
> 1. **Standard Practice in CI Testing**: Asymptotic approaches are a cornerstone of CI testing. Many well-established tests, including the classical Chi-square test [2], Fisher-z test [3], kernel-based tests [4] and more recent approaches [5], rely on asymptotic theory. While permutation-type tests may be constructed controlling Type I error in some special cases [1], there are currently no non-asymptotic methods available for CI tests involving hidden variables under nonparanormal models.
>
>
> 2. **Finite Sample Solution**: Recognizing the limitations inherent in asymptotic analysis, our framework can be readily extended with resampling techniques, such as the bootstrap. These methods offer a viable route to improve finite-sample performance without deviating from the core asymptotic framework.
>
> 3. **Empirical Support from Simulation Studies**: Extensive simulation studies in the literature demonstrate that asymptotic approximations yield accurate results even at moderate sample sizes (e.g., n>100 [4, 6]), supporting their practical applicability.
>
>
> > results from DCT
>
> We are sorry for the confusion. Both DCT and our method involve inferring the precision matrix, for which the standard technique of nodewise regression is employed in both methods [7,8]. This is why the derived CI test in Theorem 3.4 has the same form (though the analytical solutions $\xi$  involved are entirely different). We have properly acknowledged it.
>
> Apart from the use of nodewise regression, all other components employ entirely different techniques.
> Our aim lies in making the CI test sample-efficient in the presence of discretization. To achieve this, we cannot follow DCT, which further binarizes observations and conducts estimation and inference based on the binarized data—a limitation explicitly acknowledged in DCT (Appendix G in [9]). Instead, we reformulate the problem as an overparameterized one and leverage GMM to provide a principled solution with theoretical guarantees. The rationale and techniques are fundamentally different.
>
> We will carefully discuss the above points in the paper to avoid further confusion. Thanks for the valuable feedback.
>
> > In Equation (1), it should be clarified that m ranges from 2 to M−1,
>
> Thanks for your great advice. We have included "$m$ is an integer ranging from $2$ to $M-1$" in our revised paper.
>
> > definitions for Σ−jj and Σ−j−j−1 should be stated explicitly
>
> Thanks for your great suggestion. Kindly note that we define the notation in right part of line 99-102: "Similarly, $\mathbf X_{-j-j}$ is the submatrix of $\mathbf X$ without $j$th column and $j$th row, and...". However, your suggestion made us realize that it might be not sufficiently clear. We have included the specific definition of $\mathbf \Sigma_{-j-j}$ and $\mathbf \Sigma_{-jj}$ in Lemma 3.4 to improve the clarity.
>
> > In the two-step procedure, is there a specific reason for limiting the process to only two steps, rather than iterating further until some form of convergence in the weight matrix is achieved? or the convergence itself is not guaranteed?
>
> Thanks for your insightful question. Yes, we can iterate the procedure alternately until convergence, and convergence is indeed guaranteed. we don't adopt the iterative update since the two-step estimator is already consistent and variance-efficient, additional iterations would offer limited benefit while incurring higher computational cost.
>
> ---
>
> [1] Berrett et al., The conditional permutation test for independence while controlling for confounders.
>
> [2] Pearson, on the criterion that a given system of deviations from the probable in the case of a correlated system ...
>
> [3] Fisher, On the "Probable Error" of a Coefficient of Correlation Deduced from a Small Sample.
>
> [4] Zhang et al., Kernel-based conditional independence test and application in causal discovery
>
> [5] Azadkia & Chatterjee, A Simple Measure of Conditional Dependence
>
> [6] Gretton et al., A kernel two-sample test
>
> [7] Qiu et al., Inference on multi-level partial correlations based on multi-subject time series data
>
> [8] Chang et al., Confidence regions for entries of a large precision matrix
>
> [9] Sun et al., A conditional independence test in the presence of discretization
>
> ---
>
>  Please feel free to let us know if any part remains unclear or if you have further questions. Thank you again for your time and valuable feedback.

---

> > ### Comment · Reviewer_bfoQ · 2025-04-07
> >
> > Thanks for your helpful response. I have increased my score.

---

### Official Review · Reviewer_iPdb · 2025-03-12

**Overall Recommendation:** 3

**Summary:**

This work tackles the problem of detecting conditional independence among hidden continuous variables, while the observed variables are discrete. Specifically, the authors rely on a recent work, the Discretization-Aware CI Test (DCT), which establishes a workflow to estimate covariances between continuous variables. This workflow leads to a system of equations that the DCT uses to over-identify the covariance. To address the issue of over-identification, the authors propose using a Generalized Method of Moments (GMM) to leverage all equations constructed with various discretization boundaries to acquire an accurate estimate of the covariance. Subsequently, nodewise regression is used, combining the covariance estimates to determine the conditional independence between variables.

**Claims And Evidence:**

Most of the claims presented are clear, except for Theorem 3.5, which is critical to demonstrating why the DCT-GMM approach is superior to the standalone DCT method. It would be beneficial if the authors could provide additional details about Theorem 3.3 in the main content to enhance understanding and support the claims made.

**Essential References Not Discussed:**

I am not aware of.

**Experimental Designs Or Analyses:**

I have reviewed the experiments, and they are generally convincing. However, I am a bit concerned about the experimental settings for the application in causal discovery. The parameters described in lines 407 to 410 on the left differ from those used in the original DCT work.

**Methods And Evaluation Criteria:**

The evaluation criteria include Type I and Type II errors, F1 Score, precision, recall, etc., which makes sense for evaluating the proposed method.

**Other Comments Or Suggestions:**

Please see the weaknesses section.

**Other Strengths And Weaknesses:**

I have noticed substantial overlap between lines 91 to 102 and lines 297 to 304 on the right-hand column with the corresponding sections of the DCT work. Please rephrase these sections to ensure your work is original.

**Questions For Authors:**

What are the benefits of using nodewise regression rather than constructing and inverting a covariance matrix? Is it because the inversion is expensive?

**Relation To Broader Scientific Literature:**

I believe the proposed method could prove particularly useful in digital data settings, where data is often discretized for storage in devices.

**Theoretical Claims:**

I did not check.

---

> ### Author Rebuttal · Authors · 2025-03-31
>
> > Most of the claims presented are clear, except for Theorem 3.5.
>
> Thank you for this construction comments.  We followed your comment and carefully revised Theorem 3.5 to make it more intutive and clear. Note that the formal theorem demonstrating the superiority of DCT-GMM over DCT is provided in Appendix G. We avoid to include it in the main body of the submission because presenting the claim requires a detailed introduction to DCT, which may detract from the overall flow of the main text.
>
> > It would be beneficial if the authors could provide additional details about Theorem 3.3 in the main content
>
> Thank you for your thoughtful feedback. We believe you are referring to Lemma 3.3, which is intended to illustrate a property of nodewise regression that supports the derivation of the CI test. Inspired by your comments, we have added sentences to clarify the motivation behind using nodewise regression in this context:
>
> " While Theorem 3.1 and Lemma 3.2 effectively address the independence test, they do not directly resolve the CI testing problem. A seemingly straightforward approach is to invert the estimated covariance matrix; however, this does not provide a valid solution for inference."
>
> > The parameters described in lines 407 to 410 on the left differ from those used in the original DCT work.
>
> Thanks for your careful review.  Since DCT is the only method specifically designed to handle discretization scenario, we generally followed their experimental setup for fair comparison. However, our parameter setting for the experiment is **more challenging**, involving **fewer samples** and **more variables** to validate the superiority of DCT-GMM.
> We summarize the comparison of causal discovery of DCT paper and our own:
>
> | Setting        | DCT                               | DCT-GMM                       |
> | -------------- | --------------------------------- | ----------------------------- |
> | varying sample | $p=8, n=(500, 1000, 5000, 10000)$ | $p=10, n=(100,500,1000,2000)$ |
> | varying nodes  | $n=5000, p=(4,6,8,10)$            | $n=2000, p=(4,6,8,12)$        |
>
> > overlap between lines 91 to 102 and lines 297 to 304 on the right-hand column with the corresponding sections of the DCT work.
>
> We appreciate the reviewer for the careful review. We have rephrased the related work part (line 91 to 102) with citing additional references [1, 2, 3] (thank again to reviewer NWpp for his great suggestion).
>
> For the experiment part (line 297 to 304), our intention is to follow the previous work DCT. Given your concern, we have rephrased it as
> "
> In the first part, we evaluate the Type I and Type II errors of DCT-GMM across various scenarios, comparing it with baseline methods including DCT (Sun et al., 2024), the Fisher-z test (Fisher, 1921), and the Chi-square test (F.R.S., 2009).  In the second part, we assess the performance of DCT-GMM on causal discovery tasks and compare with same baseline methods.  In the third part, we directly compare DCT-GMM and DCT to empirically validate Theorem 3.5.
> "
>
> > Benefits of using nodewise regression rather than inverting a covariance matrix?
>
> Thanks for your insightful question. The problem with inverting the covariance matrix is that **it does not address the issue of inference**, i.e., **deriving the distribution** of $\hat \omega_{jk} - \omega_{jk}^*$.
>
> While the GMM solves the estimation of covariance $\mathbf{\hat{\Sigma}}$ effectively, and we can directly invert it and obtain $\mathbf{\hat{\Omega}}$ , whose entry $\hat \omega_{jk}$ captures the relation of $X_j$ with $X_k$ conditioning on all other variables. The problem is, it does not provide a tractable way to infer the distribution $\hat \omega_{jk} - \omega_{jk}^*$.
> To address this, we adopt a standard technique---nodewise regression [4, 5]. This allows us to:
> 	1. Show $\beta_{j,k}$ acts as an effective surrogate of $\omega_{jk}$, thereby the wanted distribution transfers from $\hat \omega_{jk} - \omega_{jk}^*$  to $\hat \beta_{j,k} - \beta_{j,k}^*$;
> 	2. Express $\hat \beta_{j,k}-\beta_{j,k}^*$ as linear combination of the known distribution $\hat{\mathbf{\Sigma}} - \mathbf{\Sigma}^*$ as equation~(11).
>
> [1] Li S, et al. K-nearest-neighbor local sampling based conditional independence testing[J]. Advances in Neural Information Processing Systems, 2023.
>
> [2] Cai Z, et al. A distribution free conditional independence test with applications to causal discovery[J]. Journal of Machine Learning Research, 2022.
>
> [3] Kim I, et al. Local permutation tests for conditional independence[J]. The Annals of Statistics, 2022.
>
> [4] Qiu, Yumou, et al. "Inference on multi-level partial correlations based on multi-subject time series data." Journal of the American Statistical Association, 2022.
>
> [5] Chang J, et al. Confidence regions for entries of a large precision matrix[J]. Journal of Econometrics, 2018.

---

### Official Review · Reviewer_NWpp · 2025-03-14

**Overall Recommendation:** 4

**Summary:**

The paper proposes a conditional independence (CI) test for testing CI relations in discretized data. It does not rely on binarizing the data to infer the CI relations between latent variables. The paper argues that it does not need to rely on binarization like the previous work to establish correct CI relations between the latent continuous variables. It addresses an issue known as over-identifying restriction problem by using Generalized Method of Moments. The authors derive the test statistic and establish its asymptotic distribution. The key limitation of this work is that it assumes all latent continuous variables follow a multivariate normal distribution.

**Claims And Evidence:**

Yes.

**Essential References Not Discussed:**

I think the paper should cite the recent work on conditional independence tests though these tests are not primarily designed for the exact same setup. I think at least citing CI tests that work with discrete data is a fair game e.g. [1]

Reference:

- [1] Li, Shuai, et al. "K-nearest-neighbor local sampling based conditional independence testing." Advances in Neural Information Processing Systems 36 (2023): 23321-23344.

**Experimental Designs Or Analyses:**

Yes. Originally, I thought the paper intentionally avoided many recent advanced baselines. However, I later understood that it is not needed because of the neat idea that the conditional independence among the original continuous variables may be wrongly tested as conditional dependence due to discretization. With faithfulness, it makes sense. The experiment demonstrates the contributions of the proposed CI test, which is more sample efficient than the previous work called DCT and outperforms it.

However, I do not follow how the authors came up with the ground truth graph to verify their test performance in the real-world experiment. I tried to look up the reference, but I don't see anything describing the ground truth on the subset of variables.

**Methods And Evaluation Criteria:**

Yes, the paper uses F1 score, precision, and recall of the skeleton to evaluate the effectiveness of the proposed CI test on learning conditional independence. The authors also show the robustness of the test based on a range of sample sizes and graph sizes. They further test the algorithm on denser graphs and a real-world experiment.

**Other Comments Or Suggestions:**

- inferring \tilde{X_{1}} being conditionally dependent of  \tilde{X_{3}} given  \tilde{X_{2}} is via faithfulness assumption, not causal Markov condition.
- Minor typo in figure 3b: “Numb”
- Lines 964-968: DCTG-> DCT-GMM. Also, the algorithm is named DCTG in the appendix, but the main paper refers to it as DCT-GMM.
- It would be better to describe how the DAG structures are generated in the experiment.
- I think it will be interesting how robust the proposed CI test is when the unobserved continuous variables do not follow a normal distribution and run the same comparison with the baselines.

**Other Strengths And Weaknesses:**

Strengths:

- The paper is quite well-written.

- The theoretical claims are logically sound. I appreciate the detailed explanation for the proofs in section F.

- The experiment supports the contributions and claims of the paper. Both synthetic and real-world data are used.

Weaknesses:

- Some sentences are difficult to understand .e.g, "the proportion of both observed variables exceeding their means reflects the underlying covariance, solved using a single equation.”

- The practical significance of the proposed CI tests is quite limited. Although the previous work (Sun et al. 2024) has been published in ICLR'25, I find it difficult to come up with practical scenarios where one knowingly discretizes some continuous variables, and those continuous variables become unobserved. This is further questioned, especially when it assumes normality for those unobserved continuous variables.

- The paper uses results heavily from (Sun et al. 2024) to derive the CI tests via nodewise regression. The presentation structure e.g., experiments, also closely follows Sun et al. 2024 (see section 3 and the rest in Sun et al. 2024).

**Questions For Authors:**

1. In Figure 3b, does ‘Fisherz’ mean that the authors directly apply the Fisher z test to the discretized data? If so, why is that appropriate?
2. How do the authors obtain the ground truth for the real-world experiment to verify the experimental result, especially for the selected three variables?
3. Can the authors further give some practical scenarios where some variables are discretized and are known to be both continuous and unobserved?
4. Can the authors test on an experiment where the unobserved continuous variables do not follow a normal distribution?

I will raise my score if the authors address my questions well.

**Relation To Broader Scientific Literature:**

The paper aims to solve the overidentification issue (see lines 184-192) found in one recent related work called DCT (Sun et al. 2024) without the need of binarization like DCT. The key contribution of this paper is the content starting from line 197 all the way to proving Theorem 3.1 (con)and Lemma 3.2 along with Theorem 3.5. Theorem 3.1 establishes the asymptotic normality of the covariance estimator derived via GMM.  Lemma 3.2 further claims that choosing an optimal weighting matrix (namely, one converging to the inverse of the covariance of the moment functions) reduces the asymptotic variance compared to a one-step estimator.  Then, the authors follow the same framework of using nodewise regression (Sun et al. 2024) to derive the CI tests. The authors also argue that DCT-GMM, the proposed CI test, achieves lower variance than DCT because it leverages additional valid moment conditions via Theorem 3.5.

**Theoretical Claims:**

Yes. I have checked sections F and G. I don’t see any issue.

---

> ### Author Rebuttal · Authors · 2025-03-31
>
> > Some practical scenarios
>
> Thank you for your valuable question. We appreciate the chance to highlight the **common and often unavoidable discretization** due to practical measurement constraints.  In principle, **any variables measuring "degree" or "intensity"**(e.g., happiness, severity) are inherently continuous but often recorded in discrete form. For example
>
> - In medical diagnostics, the **severity of cancer** is commonly categorized into stages like _"Stage I"_ or _"Stage II"_, not because the disease progresses in discrete jumps, but because there is no equipment that can exactly quantify its severity on a truly continuous scale.
>
> - Similarly, in questionnaires—including the real-world dataset used in our paper—latent continuous variables such as **level of depression** are often measured using Likert scales (e.g., 1 to 5) for fast assessment and clinical usability.
>
> We hope those practical scenarios can clarifies its practical significance of DCT-GMM.
>
> > Groundtruth of real-world dataset
>
> Thanks for your question. You're right—there is no ground truth in the Big Five Personality Dataset. We followed the setup from DCT. Despite the absence of ground truth for reference, the results of the discretization-aware CI tests (DCT and DCT-GMM) appear more reasonable. Specifically, the conclusion that $N4 \perp N10 \mid N3$, is intuitively plausible---since N10 represents "I often feel blue" is a stronger indicator of mood and should naturally subsume the information in N4 ("I seldom feel blue").
>
> At the same time, we would greatly appreciate it if the reviewer could suggest any datasets that align well with our setting.
>
> > Normal assumption
>
> Thank you for this very insightful question. Please kindly refer to our **response~1 to Reviewer 9WLL** for a detailed explanation. Due to the 5000-character limit, we were unable to include the full response here. We sincerely apologize for the inconvenience.
>
> > Citing CI tests that work with discrete data...
>
> Thanks for your great suggestion. We have included the suggested reference [1] and other recent CI tests for discrete variables in our revised version [2,3].
>
> > Sentences are difficult to understand. e.g, the proportion...
>
> Thank you for the helpful comment. We have followed your comment and revised it to: “The empirical probability of observed discrete pairs reflects the covariance of the underlying continuous variables.”
>
> > Use heavily from DCT of nodewise regression
>
> Thank you for discussing the connection between DCT and ours. Nodewise regression is a standard technique for addressing the inference problem of the precision matrix [4,5]. Both DCT and our method involve the inference of the precision matrix.  By using the nodewise regression in nonparanormal model, the derived CI test in Thoerem 3.4 would have the same form (but the analytical solutions involved are entirely different).  We have properly acknowledged this in the main text and references.
>
> Moreover, it is worth noting that apart from the use of nodewise regression, all other components employ entirely different techniques.
> Our unique contribution lies in making the CI test sample-efficient in the presence of discretization. To achieve this, we cannot follow DCT, which further binarizes observations and conducts estimation and inference based on the binarized data—a limitation explicitly acknowledged in DCT (Appendix G in [6]). Instead, we reformulate the problem as an overparameterized one and leverage GMM to provide a principled solution with theoretical guarantees. The rationale and techniques are fundamentally different.
>
>
> > - DAG generation
>
> Kindly refer to lines 380–382, where we state that the DAG is generated using the BP model.
>
> > "Fisherz" meaning..
>
> Thank you for your insightful question. Yes, "Fisherz" refers to directly applying the Fisher z test to discretized data. Our goal is twofold:
>
> 1. To compare it with the Fisher z test on the original continuous data and highlight how discretization distorts causal discovery.
> 2. To show that even if users know the variables are inherently continuous, directly applying a CI test designed for continuous data—such as the Fisher z test—on discretized values can significantly degrade the resulting causal graph.
>
>
> [1] Li et al., K-nearest-neighbor local sampling based conditional independence testing
>
> [2] Cai et al., A distribution free conditional independence test with applications to causal discovery
>
> [3] Kim et al., Local permutation tests for conditional independence
>
> [4] Qiu et al., Inference on multi-level partial correlations based on multi-subject time series data
>
> [5] Chang et al., Confidence regions for entries of a large precision matrix
>
> [6] Sun et al., A conditional independence test in the presence of discretization
>
> ---
>
> We sincerely thank the reviewer for the thorough and constructive review, which has greatly improved our paper. Please let us know if there are any further questions.

---

> > ### Comment · Reviewer_NWpp · 2025-04-04
> >
> > I thank the authors for the response and the additional experiments. I have read other reviewers' comments and responses. Overall, I think the paper makes a decent theoretical contribution and it's evident in the experimental results. I am happy to raise my score to support this paper to be accepted.

---

> > > ### Author Response · Authors · 2025-04-04
> > >
> > > We sincerely appreciate your thoughtful review and for raising your score. Your constructive feedback has greatly improved our paper. Thank you for taking the time to carefully consider our rebuttal and for acknowledging the contributions of our research.

---

### Decision · Program_Chairs · 2025-05-01

**Decision:**

Accept (poster)

**Comment:**

The paper presents a method for conditional independence testing where the variables are continuous but discretized during measurement, often seen in social science, without binarization. It provides sound asymptotic analysis and experiments on both synthetic datasets indicating improvements over previous methods. However, like many previous works, it assumes that the variables follow a multivariate Gaussian distribution.